# Site-selected in situ polymerization for living cell surface engineering

Yihong Zhong[1], Lijia Xu[1], Chen Yang[1], Le Xu[2], Guyu Wang[1], Yuna Guo[1], Songtao Cheng[1], Xiao Tian [3], Changjiang Wang [3], Ran Xie [3,4], Xiaojian Wang [2] ✉, Lin Ding [1,4] ✉ & Huangxian Ju[1]

The construction of polymer-based mimicry on cell surface to manipulate cell behaviors and functions offers promising prospects in the field of biotechnology and cell therapy. However, precise control of polymer grafting sites is essential to successful implementation of biomimicry and functional modulation, which has been overlooked by most current research. Herein, we report a biological site-selected, in situ controlled radical polymerization platform for living cell surface engineering. The method utilizes metabolic labeling techniques to confine the growth sites of polymers and designs a Fenton-RAFT polymerization technique with cytocompatibility. Polymers grown at different sites (glycans, proteins, lipids) have different membrane retention time and exhibit differential effects on the recognition behaviors of cellular glycans. Of particular importance is the achievement of in situ copolymerization of glycomonomers on the outermost natural glycan sites of cell membrane, building a biomimetic glycocalyx with distinct recognition properties.

Live cell surface engineering is the use of chemical, biological, and/or material science methods to tailor the landscape of the cell surface and modulate cell function while maintaining cellular activity[1-3]. Artificial polymers are not only the most readily available models for mimicking the structure and function of natural biomacromolecules, but also can present biophysical and biochemical cues that natural macromolecules are difficult to present, through bottom-up design[4-6]. Engineering living mammalian cells with artificial polymers can not only provide rich models for understanding the organization of complex biological systems and their physiological functions, but also advance the engineering of human cells and tissues[7-9], with promising applications in transplantation[10], personalized cell therapy[11], regenerative medicine[12], drug delivery[13], and biosensing/bioimaging[14].

The most common methods of cell surface polymer engineering are to attach synthetic polymers to the cell membrane by lipid insertion[7,15-18], electrostatic adsorption[19], and covalent bonding[20,21]. For example, Bertozzi's group inserted the conjugates of glycopolymers and phospholipids onto the surface of cancer cells to reveal the relationship between hypersialylation and immunoprotection[22]. However, the steric hindrance of both the natural cellular macromolecules (e.g., proteins, glycans) and artificial polymers can lead to limited grafting efficacy[5,23]. Compared to synthetic polymers, small monomers can break through the spatial barrier of natural macromolecules and reach the cell surface more easily. In a pioneering work, Hawker's group inserted chain transfer agents (CTA) into Jurkat T-cell membranes in a non-covalent manner to initiate controlled radical polymerization (CRP) from the membrane[24], providing a solution to the steric hindrance of cell surface polymer modification.

However, most of the current polymer-based cell surface engineering approaches overlook the effect of the graft site on the engineered cells. Considering the pronounced spatial heterogeneity and complex molecular composition of cell surfaces[25], there is an urgent

[1]State Key Laboratory of Analytical Chemistry for Life Science, School of Chemistry and Chemical Engineering, Nanjing University, Nanjing 210023, China. [2]Institute of Advanced Synthesis, School of Chemistry and Molecular Engineering, Nanjing Tech University, Nanjing 211816, China. [3]State Key Laboratory of Coordination Chemistry, School of Chemistry and Chemical Engineering, Nanjing University, Nanjing 210023, China. [4]Chemistry and Biomedicine Innovation Center (ChemBIC), Nanjing University, Nanjing 210023, China. ✉e-mail: ias_xjwang@njtech.edu.cn; dinglin@nju.edu.cn

need to develop techniques that can install polymers to specific cell sites in order to enable these polymers to perform pre-defined functions accurately and precisely. We aim to achieve polymer localization by controlling the grafting position of CTA on the cell surface. Inspired by the bioorthogonal chemical reporter technology, where metabolic labeling reagents with bioorthogonal groups can selectively label biological sites by hijacking specific intracellular synthetic pathways[26,27], we combine metabolic labeling and click reactions for CTA installation. The covalent modification to selected types of biomolecules avoids unnecessary interference with the cells and provides the advantage of a more stable linkage and longer retention time of the grown polymer on the cell membrane. In the meantime, in situ polymerization on the surface of living cells is very difficult: (1) cell viability is difficult to maintain; (2) the cell surface environment is highly complex, which often leads to the occurrence of the polymerization inhibition[24,28]. Therefore, we take advantage of the redox-responsive Fenton-reversible addition-fragmentation chain-transfer (RAFT) polymerization system with relatively mild reagents, aqueous and room temperature reaction conditions, and a very short induction time[29–32] to establish a living cell-compatible CRP platform.

We herein develop a redox-initiated CRP strategy that triggers polymerization at selected sites on the surface of living cells (Fig. 1a), namely Site-Selected in situ Polymerization (SSP). Generally, bioorthogonal groups are introduced at different sites (glycans, proteins, lipids) of cell membrane via metabolic labeling for CTA mounting, followed by Fenton-RAFT polymerization to build polymers in situ at selected sites. Using Jurkat T cells as a model, the cytoskeleton and various physiological functions are well preserved after polymerization. Polymers grown at different sites show differential effects on cells, allowing us to confer resistance to lectin-induced apoptosis on T cells by selecting growth sites. We also grow copolymers of different compositions in situ on living cells to mediate the longitudinal assembly of ligands at selected sites. Of particular interest is the copolymerization of glycan-modified monomers at the end of the cell's natural glycan chains, which constructs a biomimetic glycocalyx on living cells. The introduced glycans can reshape the recognition functions of the cells, and can be further remodeled in situ. This work provides a versatile platform for studying and regulating cellular interactions with extracellular substances.

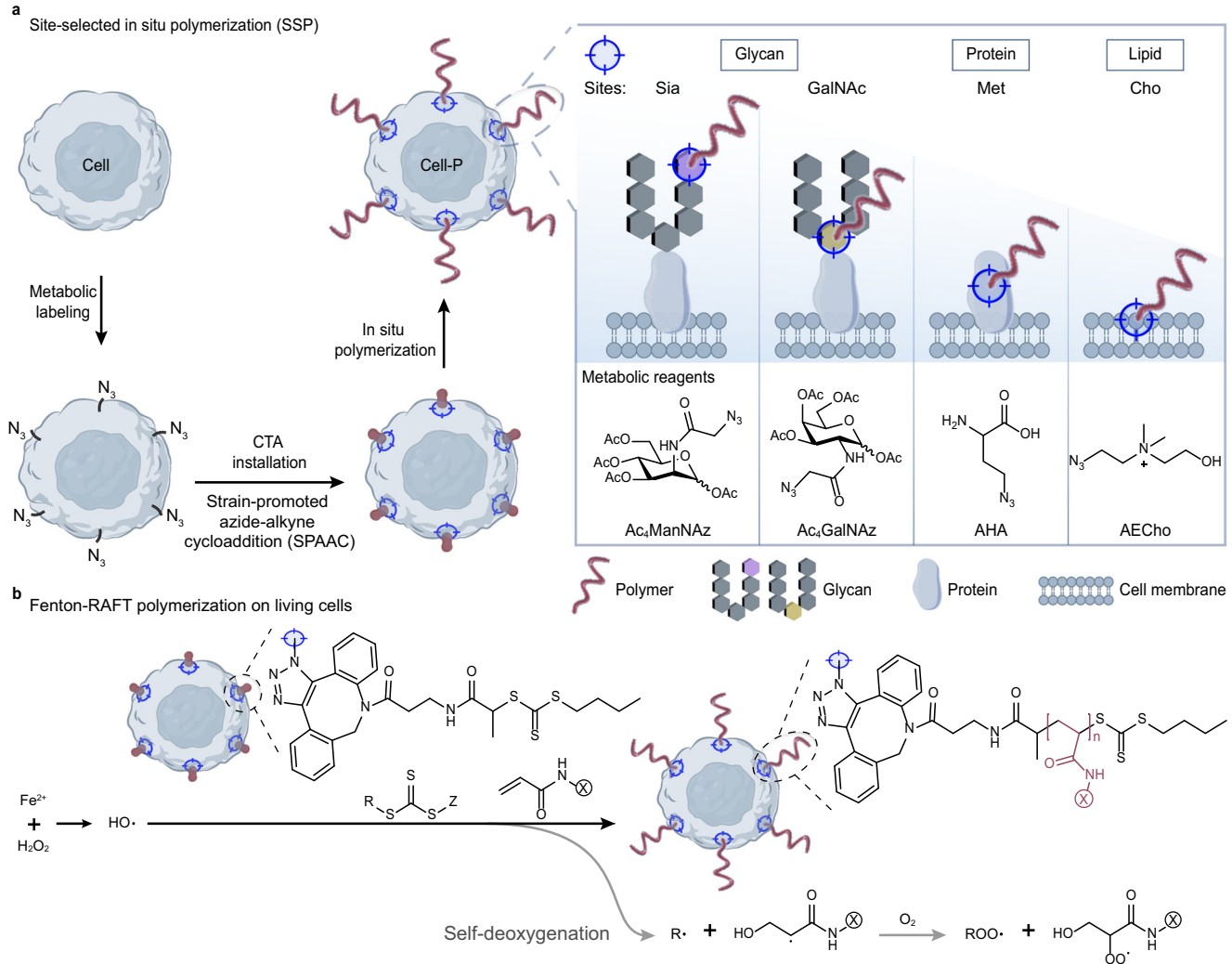

**Fig. 1 | Schematic depicting the site-selected in situ polymerization (SSP) on the surface of living cells. a** Confinement of the chain transfer agent (CTA) site by metabolic labeling allows polymerization to occur at selected glycan (sialic acid, Sia; mucin-type O-linked N-acetylgalactosamine, GalNAc), protein (methionine, Met) and lipid (choline, Cho) sites on the cell surface. **b** Proposed mechanism of the living cell surface-initiated, self-deoxygenating Fenton-RAFT polymerization reaction. Live cells with installed CTA were mixed with a solution containing monomer, free CTA, and a small amount of H₂O₂. The polymerization reaction was initiated by the addition of ammonium ferrous sulfate (which provides Fe²⁺).

## Results and discussion

### Design of cytocompatible redox-initiated CRP method

Mammalian cells are highly susceptible to death by physical or chemical damage factors, hypoxia, and nutritional deficiencies. In situ polymerization on mammalian cell surfaces requires a comprehensive assessment of polymerization efficiency and the effects on cellular activity. Fenton-RAFT is a process of controlled radical polymerization of vinyl monomers by using highly reactive hydroxyl radicals generated by the reaction of Fenton reagents (i.e., $Fe^{2+}$ and $H_2O_2$) in the presence of a suitable RAFT CTA. The highly efficient production of hydroxyl radicals makes the polymerization reaction fast[30]; on the other hand, it makes the system inherently capable of certain chemical deoxygenation[32] (Fig. 1b), which helps to solve the problem of oxygen-induced polymerization inhibition in cell suspensions.

We chose Jurkat T cells as a model, using N-(2-hydroxypropyl) methacrylamide (HPMA) as polymerization monomer[33] and 2-{[(butylsulfanyl)carbonothioyl]sulfanyl}propanoic acid (BTPA) as CTA[34,35], with 1.5 mM $H_2O_2$ and 0.5 mM $Fe^{2+}$ [30]. These individual polymerization components had little effect on cell viability (Supplementary Fig. 1). We first investigated the polymerization conditions in physiological buffers in the presence of cells and found that the key point for successful polymerization on living cells is the selection of polymerization medium. After testing various buffer media, for example HEPES (Supplementary Fig. 2), we determined that the optimal polymerization system was RPMI-1640 medium + B27 supplement minus AO (pH = 6.6–6.9, as acidic media with pH close to 7 is conducive to the reaction[29]). Chain growth properties and excellent polymerization control remained in the polymerization medium (Supplementary Fig. 3). The percentage of living cells were in the range of 98.3-98.5% (Supplementary Fig. 4a, b) and the proliferation ability were largely maintained (Supplementary Fig. 4c) when the polymerization time was within 10 and 5 min, respectively; the monomer conversion increased slightly by increasing polymerization time from 0 to 10 min, in the range of 14-24% (Supplementary Fig. 4d). Hypoxic conditions did not affect cell viability, proliferation, and apoptosis levels (Supplementary Fig. 5). To avoid damage to cells and maintain control of polymerization, high monomer conversion is undesirable and should preferably be limited to 30%[24]. Therefore, the polymerization time was set to 2 min. We finally determined the composition of the solution polymerization system as $[monomer]_0:[CTA]_0:[H_2O_2]_0:[Fe^{2+}]_0 = 80:1:0.3:0.1$. The polymerization reaction was initiated by adding a small amount of ammonium ferrous sulfate, and terminated by exposing the reaction system to air and diluting the solution, and the conversion could reach more than 20%.

### Fenton-RAFT polymerization at selected sites on living cells

To precisely control the growth sites of the polymers and to explore whether the polymers could be engineered in a covalent manner to mammalian cell membranes, we used metabolic labeling techniques to introduce clickable tags (azide), at selected cell sites. By coupling BTPA with dibenzoazacyclooctyne (DBCO), we prepared DBCO-BTPA as a CTA for covalently anchoring to the metabolically installed azides. We chose tetraacylated N-azidoacetylmannosamine (Ac4ManNAz), tetraacylated N-azidoacetylgalactosamine (Ac4GalNAz), azidohomoalanine (AHA) and 1-azidoethyl-choline (AECho) as metabolic reagents, to introduce azide group at selected sites on cell surface: Sia (sialic acid, glycan site)[36], mucin-type O-GalNAc (N-acetylgalactosamine, glycan site)[37], Met (methionine, protein site)[38], and Cho (choline, lipid site)[39], respectively (Fig. 1a). The conversion of installed azide tags to CTA at the cell surface is a crucial parameter for SSP process. A reaction temperature at 4 °C is preferred due to the hydrophobic nature of DBCO-BTPA structure. The cell viability can be maintained for 1-h click reaction with 40 μM and 100 μM DBCO-BTPA concentrations (azido-sialic acid labeled cells, highest extent of azide installation, Supplementary Figs. 1b and 6). The labeling ratios corresponding to 40 and 100 μM DBCO-BTPA were 26.7% and 37.3%, respectively (Supplementary Fig. 7). These data suggest that increasing DBCO-BTPA concentration is conducive to obtaining higher labeling ratio on the cell surface. However, due to the sensitivity of mammalian cells to the external environment, the conditions of the click reaction need to be carefully optimized.

To facilitate the CTA amount regulation on cell surface for comparing the engineering outcomes, we set the concentration of DBCO-BTPA as 40 μM and adjusted the concentrations of metabolic reagents. The amount of CTA on cellular GalNAc, Met, and Cho sites was adjusted to be consistent ($1.4 \times 10^8$ per cell), while for Sia site, the amount was $4.0 \times 10^8$ per cell, 2.9 times that of the other three conditions (as the model with the highest grafting density) (Supplementary Fig. 8).

To four suspensions of Jurkat T cells with CTA installed at different sites, we performed Fenton-RAFT polymerization at $[HPMA]_0:[CTA]_0:[H_2O_2]_0:[Fe^{2+}]_0 = 80:1:0.3:0.1$ to obtain PolyHPMA (PHPMA)-grafted Jurkat T cells (Cell-P): $Cell_{Sia}$-P, $Cell_{GalNAc}$-P, $Cell_{Met}$-P and $Cell_{Cho}$-P (the subscript of Cell indicates the selected site). We collected the reaction solutions after polymerization for ¹H-NMR characterization[24], and calculated the monomer conversion of ~27%, ~27%, ~26%, and ~25% for the four systems, respectively (Supplementary Fig. 9), demonstrating the occurrence of polymerization reaction. In contrast to native cells, the scanning electron microscope (SEM) images of Cell-P showed a rough and fluffy surface[40] (Fig. 2a). The morphology of cells without CTA after the polymerization operation was consistent with that of native cells, indicating that the polymers in solution were not adsorbed on the cell surface. These results confirm the successful growth of the polymer on the cell surface.

To directly characterize the polymers grown on the cell surface, we synthesized the chain transfer agent DBCO-SS-BTPA, composed of a cleavable disulfide bond, and attached it to the azide tag on the cell surface Sia. After polymerization, the product was separated from the cells by cleavage with tris(2-carboxyethyl)phosphine hydrochloride (TCEP). The ¹H-NMR characteristic peaks of PHPMA cleaved from the cells corresponded one-to-one with PHPMA prepared by solution polymerization (Fig. 2b), further proving successful in situ polymerization. Gel permeation chromatography (GPC) characterization of PHPMA grown on the cell surface showed PDI of 1.37, confirming the controlled nature of the SSP reaction (Supplementary Fig. 10).

### In situ copolymerization of functional monomers at selected sites on living cells

The convenience of modulating polymer composition will greatly enhance the flexibility and ease with which researchers can remodel cell membranes and modulate cell function. Therefore, we next explored the possibility to copolymerize functional polymers on living cells. After confirming the feasibility on fixed cells (Supplementary Fig. 11), we copolymerized HPMA and acrylamide-poly (ethylene glycol)4-biotin (AA-PEG4-biotin) on living Jurkat T cells to produce poly (HPMA-co-AA-PEG4-biotin)-engineered Jurkat T cells (using Sia as the graft site, $Cell_{Sia}$-$P^{biotin}$). CLSM imaging of $Cell_{Sia}$-$P^{biotin}$ after streptavidin-cyanine3 (SA-Cy3) staining revealed a strong clustered fluorescent signal on the surface of the living cells (Fig. 2c), which may be one of the characteristics of a polymer grown in situ on the cell surface, while there was negligible fluorescence inside the cells. To demonstrate that the fluorescence was indeed coming from the polymer grown on the cell surface, we also performed three sets of control experiments and found that SA-Cy3 was not adsorbed to the cell surface whether CTA was installed or not. In particular, when the Sia site was not modified with CTA, the monomer in solution and the resulting polymer did not adsorb to the cell surface to give a fluorescent signal, even when we initiated a polymerization reaction in solution (Fig. 2c and supplementary Fig. 12). The copolymerization strategy allowed us to easily confirm the success of in situ polymerization by observing the

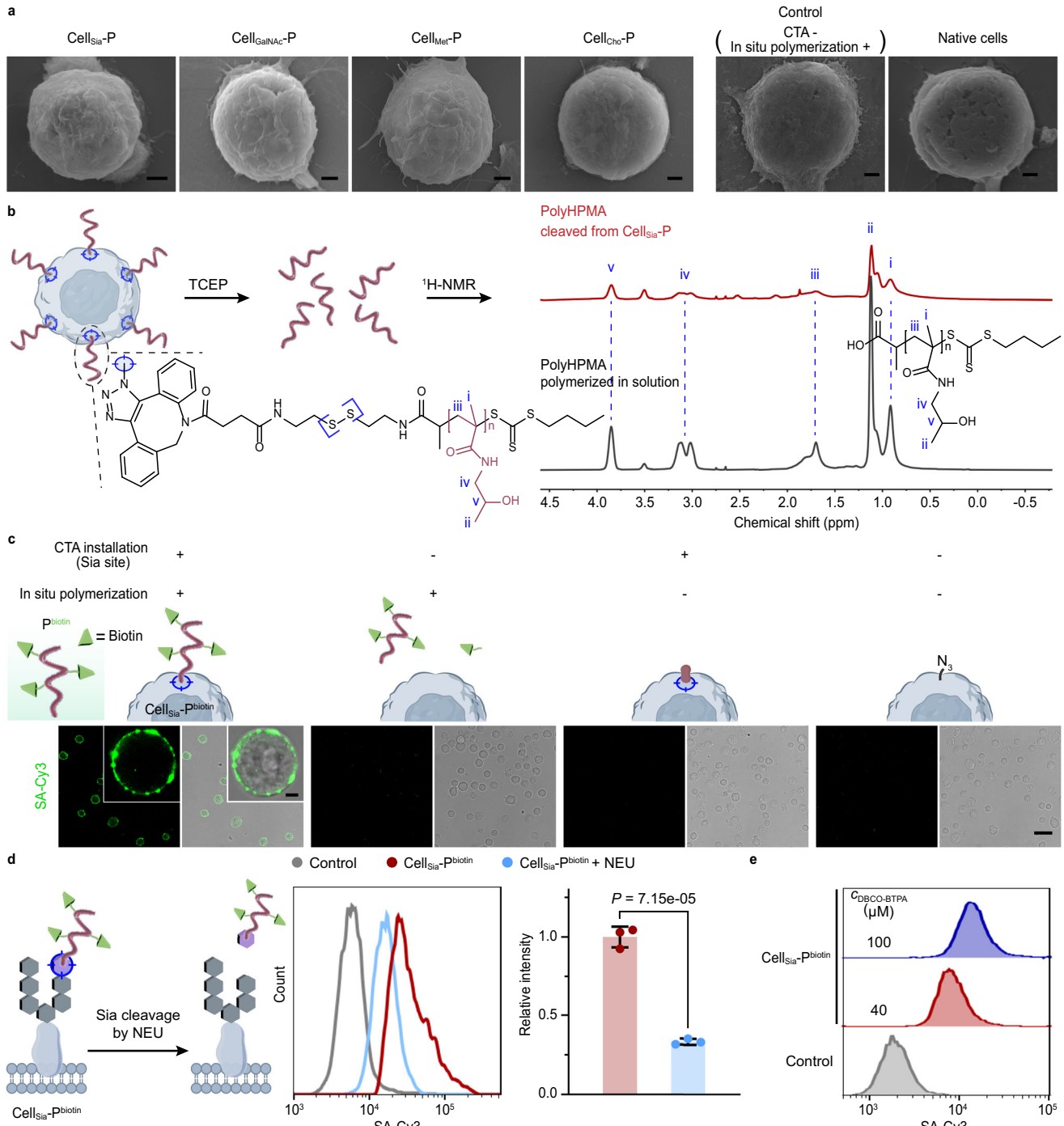

**Fig. 2 | Demonstration of the successful SSP. a** Scanning electron microscope (SEM) images of Jurkat T cells with PolyHPMA grown from different sites (Cell$_{Sia}$-P, Cell$_{GalNAc}$-P, Cell$_{Met}$-P, Cell$_{Cho}$-P). CTA-free cells undergoing polymerization operation and native cells were used as controls. Scale bars, 1 μm. **b** $^1$H nuclear magnetic resonance ($^1$H-NMR) spectra of polyHPMA collected from Cell$_{Sia}$-P by tris(2-carboxyethyl)phosphine hydrochloride (TCEP) treatment (red line, CTA: DBCO-SS-BTPA) and PolyHPMA synthesized in solution (black line, CTA: BTPA). **c** In situ copolymerization of AA-PEG$_4$-biotin and HPMA at Sia site, generating P$^{biotin}$ at cell surface. The assembly of streptavidin-cyanine3 (SA-Cy3) to P$^{biotin}$ on cell surface was visualized by confocal laser scanning microscopy (CLSM). Three control experiments were performed by omitting CTA installation and/or polymerization step. Scale bar, 25 μm. Zoomed-in images showed a single stained cell; scale bar, 3 μm.

**d** Demonstration of site-selected polymerization using Cell$_{Sia}$-P$^{biotin}$ as a model. Representative flow cytometry (FCM) histogram of Cell$_{Sia}$-P$^{biotin}$ after neuraminidase (NEU)-based Sia cleavage and SA-Cy3 staining is shown. NEU-untreated cells were analyzed for comparison. Cell$_{Sia}$-azide was used as a control. Data are shown as the mean ± SD of $n = 3$ individual experiments. Statistical differences were determined by a two-tailed unpaired Student's $t$-test. $P < 0.05$ was considered statistically significant. **e** After treatment of Cell$_{Sia}$-azide with DBCO-BTPA at 40 or 100 μM, in situ copolymerization was performed. The yielded Cell$_{Sia}$-P$^{biotin}$ was stained with SA-Cy3, followed by FCM analysis. Cell$_{Sia}$-azide was used as a control. In **a**–**c** and **e**, data are representative of three independent experiments with similar results. Source data are provided as a Source Data file.

fluorescent signal. To demonstrate that the polymer was indeed grafted from the Sia site, we treated $Cell_{Sia}$-$P^{biotin}$ with neuraminidase (NEU), a glycosidase that specifically cleaves Sia[41], and found a 66.5% reduction in the binding signal of SA-Cy3 (Fig. 2d). This percentage was comparable to the typical NEU cleavage ratio (68.6%, Supplementary Fig. 13, differences in NEU cleavage activity for different types of glycosidic linkages result in incomplete cleavage of Sia on the cell surface), demonstrating that the polymer was indeed grown at the Sia site. The degree of polymer engineering on the cell surface can be regulated by changing the amount of installed CTA: When the concentration of DBCO-BTPA was increased from 40 to 100 µM, the SA-Cy3 staining signal of $Cell_{Sia}$-$P^{biotin}$ was significantly increased (Fig. 2e).

The growth of polymers may achieve effective longitudinal amplification of the signal of the selected sites. To demonstrate this, considering the SSP process starting from the azide tag, we copolymerized HPMA and AA-$PEG_4$-azide on living Jurkat T cells to produce poly (HPMA-co-AA-$PEG_4$-azide)-engineered Jurkat T cells ($Cell_{GalNAc}$-$P^{azide}$, $Cell_{Met}$-$P^{azide}$ and $Cell_{Cho}$-$P^{azide}$). Changes in the amount of azide on the cell surface can be reflected by the DBCO-Cy5 fluorescence signal (Fig. 3a–d): (1) A strong Cy5 fluorescence signal was observed after metabolic labeling with $Ac_4$GalNAz (or AHA, AECho); (2) Partial occupation of the azide sites after covalent anchoring of CTA on the cell surface, resulted in a weakened fluorescence. (3) After 2-min polymerization, the azide groups amplified in the longitudinal direction and the signal of Cy5 was greatly enhanced. We shortened the polymerization time to 1 min and the fluorescence intensity of the resulting polymer-grafted cells was weaker than that of the corresponding cells undergoing polymerization for 2 min, suggesting the controlled nature of the polymerization method (Supplementary Fig. 14a–d). We also achieved the copolymerization of HPMA and AA-$PEG_4$-azide on a breast cancer cell line, MCF-7 (Supplementary Fig. 15), confirming the cellular applicability of the SSP strategy.

For polymer-based engineering of cells, the dynamic distribution of polymers on the cell surface are of interest to researchers. We used DBCO-Cy5 to stain $Cell_X$-$P^{azide}$ (X represents the polymer growth site) and then incubated the cells for 24 h to investigate the retention of polymers grown at different sites on the cell membrane (Fig. 3e, f). After incubation, the polymer retention ratio (RR) ranged from 53.0 to 76.3%. The reduction of polymers on the cell membrane can be attributed to two reasons: (1) dilution of the polymers due to cell proliferation and (2) endocytosis of the polymer-modified biomolecules into the cells (fluorescent signal could be seen inside the cells) or exocytosis to the outside of the cells. The RR of the polymer on $Cell_{GalNAc}$-$P^{azide}$ was significantly higher than that on $Cell_{Met}$-$P^{azide}$ and $Cell_{Cho}$-$P^{azide}$. This trend was consistent with that of cells metabolically labeled (azide) at different sites (Cell-azide), but the difference was greater (Fig. 3f). This may be due to the greater influence of the polymer in altering the dissociation and recycling kinetics of cell membrane proteins or lipids. We also analyzed the RR of polymers on cells polymerized at different sites for 1 min (Fig. 3f and supplementary Fig. 14e). As expected, the decrease in the molecular weight resulted in reduced RR. The above results suggest that SSP provides a strategy to regulate the membrane retention time of the polymer as well as the modified biomacromolecules by selecting growth sites.

To better observe the morphology and evaluate the retention ability of the polymer on the cell membrane, we performed stimulated emission depletion (STED) super-resolved imaging of $Cell_{GalNAc}$-$P^{azide}$, $Cell_{Met}$-$P^{azide}$ and $Cell_{Cho}$-$P^{azide}$ using Click-iT™ sDIBO as the fluorescent signal molecule (Fig. 3g). The polymer was clearly visible in the STED images distributed in clusters on the cell membrane, and when the polymerization time was extended, the fluorescent clusters were larger in size and more clearly delineated compared to the background. We also incubated the stained cells for another 24 h and found that the cell surface polymers still appeared in clusters, but with weaker intensity than in the samples imaged immediately after polymerization, and the

fluorescence signal appeared inside the cells (Fig. 3h). After 24 h of incubation, cells with longer polymerization time also showed larger fluorescent clusters (Fig. 3h). For comparison, we also observed Cell-azide with STED and found that, regardless of the sites of azide tag introduction, the cell membranes showed a continuous distribution of weak fluorescence (Supplementary Fig. 16), which was very different from that of polymer-grafted cells. The fluorescence distribution in Cell-azide after 24 h of incubation was consistent with the CLSM imaging results (Fig. 3a). Considering the uniform distribution of the azide tags on cells after metabolic labeling (Supplementary Fig. 16a), as well as the uneven distribution of polymers on fixed cells undergoing an otherwise identical polymerization process (Supplementary Fig. 11), we speculate that the uneven distribution of polymers on the cell surface most likely occurs at the polymerization stage, which may be due to proximity-triggered cooperative polymerization[42].

To investigate the cell membrane stability of polymers grown from different sites, we cultured $Cell_X$-$P^{azide}$ for different periods of time, followed by DBCO-Cy5 staining and FCM analysis (Supplementary Fig. 17). The half-lives of the polymers grown from GalNAc, Met, and Cho sites were 54, 30, and 24 h, respectively, and the percentage of retention was less than 20% after 120, 48, and 36 h of culturing. These results consistently indicate that the cell membrane stability of the polymers grown from the glycan sites is significantly higher than that of polymers from the other two sites.

## Study of the biological effects of polymers in situ grown on living cells

After polymerization, the viability of Cell-P with different polymer growth sites showed insignificant statistical difference compared to native cells (99.1%) (Fig. 4a); and all four types of Cell-P exhibited proliferation ability within 72 h (Fig. 4b). $Cell_{GalNAc}$-P had similar proliferation ability as native cells. $Cell_{Met}$-P proliferated slowly at first, but the cell number was almost the same as native cells after 72 h. $Cell_{Cho}$-P proliferated slowly after 24 h. $Cell_{Sia}$-P proliferated slowly compared to native cells. We also examined the cell proliferation ability of Cell-azide and Cell-CTA. Overall, metabolic labeling had little effect on cell proliferation, while the proliferation ability of Cell-CTA slightly decreased compared to native cells, but the overall decrease was less than that of Cell-P (Supplementary Fig. 18), suggesting that the reason for the change in cell proliferation was due to the polymer growth. Polymer growth essentially did not affect the cytoskeleton of Cell-P, as confirmed by the maintained F-actin expression level[43] (Supplementary Fig. 19) and orientation[44] (Fig. 4c, d).

Considering that the in situ polymerization reaction may lead to an imbalance in intracellular reactive oxygen species (ROS) levels, we confirmed with a 2′,7′-dichlorodihydrofluorescein diacetate (DCFH-DA) probe that the polymer growth at all three sites except the Sia site did not lead to significant changes in intracellular ROS levels[45] (Supplementary Fig. 20); a slight increase in ROS levels in $Cell_{Sia}$-P was presumably related to the larger amount of CTA installed at the Sia site (2.9 times as many as other sites, Supplementary Fig. 8).

We further investigated whether Fenton-RAFT radical polymerization leads to intracellular DNA and protein damage. Using phosphorylated histone H2AX (γ-H2AX) to indicate the level of DNA damage[46], we found that only in $Cell_{Sia}$-P the damage signal was slightly increased; after incubation of $Cell_{Sia}$-P for 48 h, the DNA was largely repaired to the level of native cells (Fig. 4e and supplementary Fig. 21). The levels of the intracellular proteins caspase-3, caspase-9, and cytochrome c, which are closely related to ROS and hypoxia[47–49], were comparable to those of native cells for all types of Cell-P immediately after polymerization and after a further incubation of 48 h, respectively (Fig. 4f), thus indicating the good cytocompatibility of our polymerization method.

Metabolism is a key mechanism in the regulation of the immune system, and the fate and function of T cells are inextricably linked to

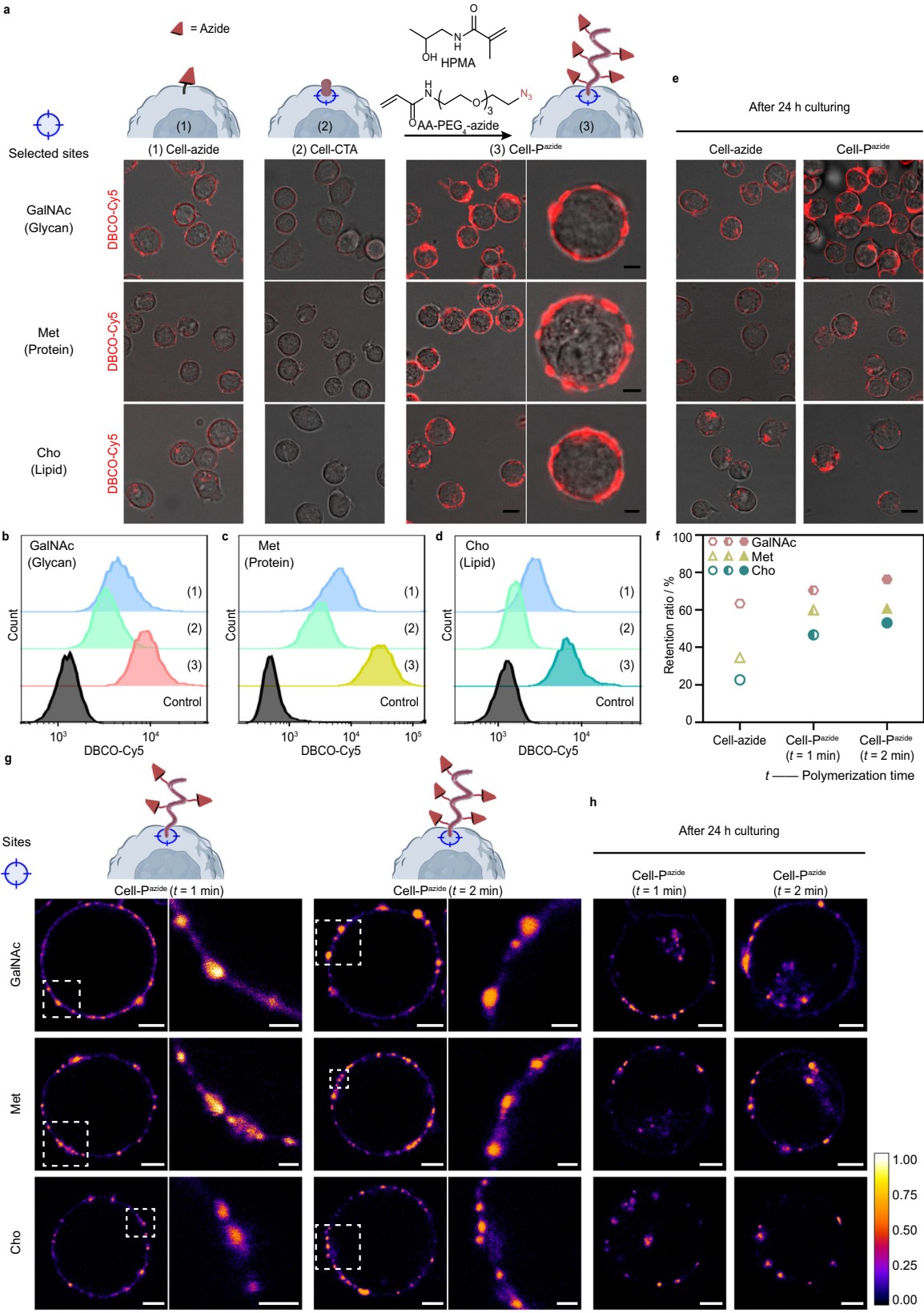

their metabolism[50]. We used Seahorse XF Cell Mito Stress Test to demonstrate that cells still had excellent basal and maximal respiration capacity after metabolic labeling and CTA installation (Supplementary Fig. 22), while after polymer growth, basal respiration levels were reduced by 19.7-35.7% and maximal respiration capacity was reduced

by 22.0-27.0% (Fig. 4g), and the metabolic level was still within an acceptable range.

Next, we examined the effect of in situ growth of polymers on the recognition and response functions of Jurkat T cells. Using CD3, a T-cell marker involved in T-cell activation[51], as the model, the antibody

**Fig. 3 | In situ copolymerization of functional monomers at selected biological sites on living cells for directing longitudinal assembly. a–d** Longitudinal amplification concept. In situ copolymerization of HPMA and AA-PEG$_4$-azide at selected sites (GalNAc, Met or Cho) to yield P$^{azide}$-engineered cells. After each step of (1)-(3), the amount of azide was assessed by dibenzoazacyclooctyne (DBCO)-Cy5 staining. The growth of P$^{azide}$ achieved efficient longitudinal amplification of the azide metabolically labeled at the selected sites. Scale bar, 8 μm. Zoomed-in images showed a single stained cell. Scale bars, 3 μm. **e, f** Evaluation of the retention time of polymers grown from different sites. **e** The metabolically-labeled cells (Cell-azide)

or P$^{azide}$-grown cells (Cell-P$^{azide}$) were staining by DBCO-Cy5, cultured for 24 h, and observed using CLSM. Scale bar, 8 μm. **f** Fluorescence intensity ratios (retention ratios) of Cell-P$^{azide}$ (with polymer grown for 1 or 2 min) or Cell-azide after 24 h of incubation versus 0 h of incubation ($n = 10$ cells per condition). **g** Stimulated emission depletion (STED) super-resolved images of single Cell-P$^{azide}$ with polymer grown (1 or 2 min) from selected sites (GalNAc, Met or Cho) (scale bar, 3 μm) and further magnified STED images (scale bar. 1 μm). Azide was stained with Click-iT™ sDIBO. **h** STED images of cells from (**g**) after 24 h of culture, scale bar, 3 μm. Data are representative of three independent experiments with similar results.

binding signals against CD3 of Cell$_{Sia}$-P and Cell$_{GalNAc}$-P were similar to those of native cells, as reflected by anti-CD3-FITC staining; and the signals of Cell$_{Met}$-P and Cell$_{Cho}$-P showed only a weak decrease (20.4% and 31.9%, respectively) (Fig. 4h, i and supplementary Fig. 23). These results suggest that the accessibility towards CD3 was maintained to some extent under the synthetic conditions. This may be due to the flexible chain polymers with limited length. However, in the cellular longitudinal direction, the approximate order of the four biological sites according to their proximity to the cellular phospholipid bilayer is: sugar, protein, and lipid (from far to near), so polymers grown from protein and lipid sites may have some effect on the accessibility of CD3 due to their closer proximity to CD3. We incubated Cell-P for another 48 h and found that the binding signal of anti-CD3 antibodies returned to levels similar to native cells (Supplementary Fig. 24), regardless of the polymerization site. The reason for this is presumably that cell proliferation reduces the density of polymers on the cells so that they can not interfere with antibody binding.

Jurkat T cells can be activated by phorbol 12-myristate 13-acetate (PMA) and ionomycin (IO)[52]. Activated T cells expressing CD69 (a measure of T lymphocyte activation) can be stained with anti-CD69-PE. Treatment of Cell$_{Sia}$-P with PMA and IO for 4 h resulted in a percentage cell activation of 91.6%, comparable to that of drug-treated native cells (Fig. 4j). In contrast, Cell-P without drug treatment did not express CD69 either immediately after polymerization or after an additional 48 h of culture (Supplementary Fig. 25), indicating that the CD69 expression in Fig. 4j was not caused by the SSP process. The above results indicate that SSP basically does not affect the biological function of Jurkat T cells. Taken together, the results show that the polymerization process has a very limited effect on the activity and function of the cells. The reasons for this may include the presence of ROS-depleting reagents in the polymerization system, the tightly controlled polymerization time (2 min), and the natural antioxidant mechanisms of the cells.

**Polymers grown at different sites have differential effects on cellular glycan recognition**

Due to the frontline sentinel position of glycans in the processes of cellular recognition and signal transduction, we focused on the effect of polymer grown from different sites on various recognition processes that glycans participate. We used galactose oxidase (GAO), sodium periodate, and lectins (Sambucus Nigra lectin (SNA), succinylated wheat germ agglutinin (S-WGA), and Jacalin) as three types of model molecules for our studies.

GAO can catalyze the oxidation of -OH at C6 position of terminal galactose (Gal) and GalNAc of cellular glycan chains to generate aldehyde groups (Fig. 5a)[53,54], which can be then fluorescently labeled by fluorescein-5-thiosemicarbazide (FTZ). To our surprise, after GAO treatment, there was no statistical difference in FTZ fluorescence between each Cell-P group with the native cell group (Fig. 5b and supplementary Fig. 26). This indicates that in situ growth of the polymer does not affect GAO oxidation, independent of the growth site. The small molecule NaIO$_4$ can selectively oxidize the neighboring hydroxyl groups of C-7 and C-8 of terminal Sia of cellular sugar chains at specific mild conditions (1 mM NaIO$_4$, neutral pH, 4 °C, 15 min,

Supplementary Fig. 27)[55], introducing an aldehyde group at the C-7 position (Fig. 5c). As shown in Fig. 5d and supplementary Fig. 28, only Cell$_{Sia}$-P showed decreased FTZ fluorescence signal (by 22.8%) compared to native cells, presumably the decrease corresponded (or partially corresponded) to the Sia sites with polymer grown. The results of the above two models collectively suggest that the polymers grown on the cell surface under our synthetic conditions allow the natural glycocalyx of cells to recognize or react with enzymes (GAO) or small molecules (NaIO$_4$), provided that the grafting site does not spatially conflict with the recognition site.

In living systems, when several keys (e.g., glycan molecules) bind together to a structure containing several locks (e.g., lectins and their assemblies), this multivalent interaction provides a significantly enhanced affinity for the biological systems[56]. This is of great importance for the biological regulatory functions of lectins, which have very weak monomeric affinity[57,58]. Therefore, we next investigated the effect of polymer growth sites on lectin recognition behavior. The three FITC-modified lectins, SNA, S-WGA, and Jacalin, can specifically recognize α2,6-Sia[59], N-acetylglucosamine (GlcNAc)[60], and galactosyl (β-1,3) N-acetylgalactosamine[61], respectively.

Unlike the cases with GAO or NaIO$_4$ treatment, polymers grown from glycan sites had a very significant effect on lectin binding: the degree of reduction ranged from 47% to 90%. In contrast, polymers grown from proteins and lipids had a weaker effect than polymers from glycan sites in all three lectin groups (Fig. 5e–h and supplementary Fig. 29). The polymer growth sites also showed differential effects on lectin-induced cell aggregation (Fig. 5e): the groups with glycans as growth sites showed significantly reduced cell aggression; while polymers grown at protein and lipid sites inhibited cell aggregation to a lesser extent than those at glycan sites.

We also investigated the effect of changing the polymer chain length on lectin binding, using S-WGA as a model (Supplementary Fig. 30). When we shortened the polymerization time to 1 min, polymers grown at the Sia or GalNAc sites had a weaker blocking effect against S-WGA than those grown for 2 min, and Cell-azide and Cell-CTA had similar lectin binding levels as native cells. Thus, we suggest that the reduction in lectin binding is due to polymer growth and not to structural changes in the glycans.

We hypothesize that the reason for these phenomena is that the cell surface-grown polymer acts as a spacer between the cell's natural glycan epitopes, making it more difficult for lectin multivalent recognition to occur (Fig. 5i). This resulted in (1) a decrease in the affinity between the cellular glycoconjugates and lectins (Fig. 5j and Supplementary Fig. 31); and (2) a decrease in the extent of cell aggregation, as observed in Fig. 5e. The effect of this spacing is closely related to the growth sites of the polymers.

Tumor cells can aid their own escape by secreting galectins to inhibit full T-cell activation, and induce T-cell growth arrest and apoptosis[62]. Based on the above results, we envisioned that the polymers grown on cell surface could help cells resist lectin-induced apoptosis (Fig. 6a). Using S-WGA treatment as a model, the apoptosis ratio of native cells reached 79.1% (20 μg/ml S-WGA) and 90.5% (40 μg/ml S-WGA), respectively (Fig. 6b, c and Supplementary Fig. 32). In contrast, the apoptosis ratio was significantly lower (not

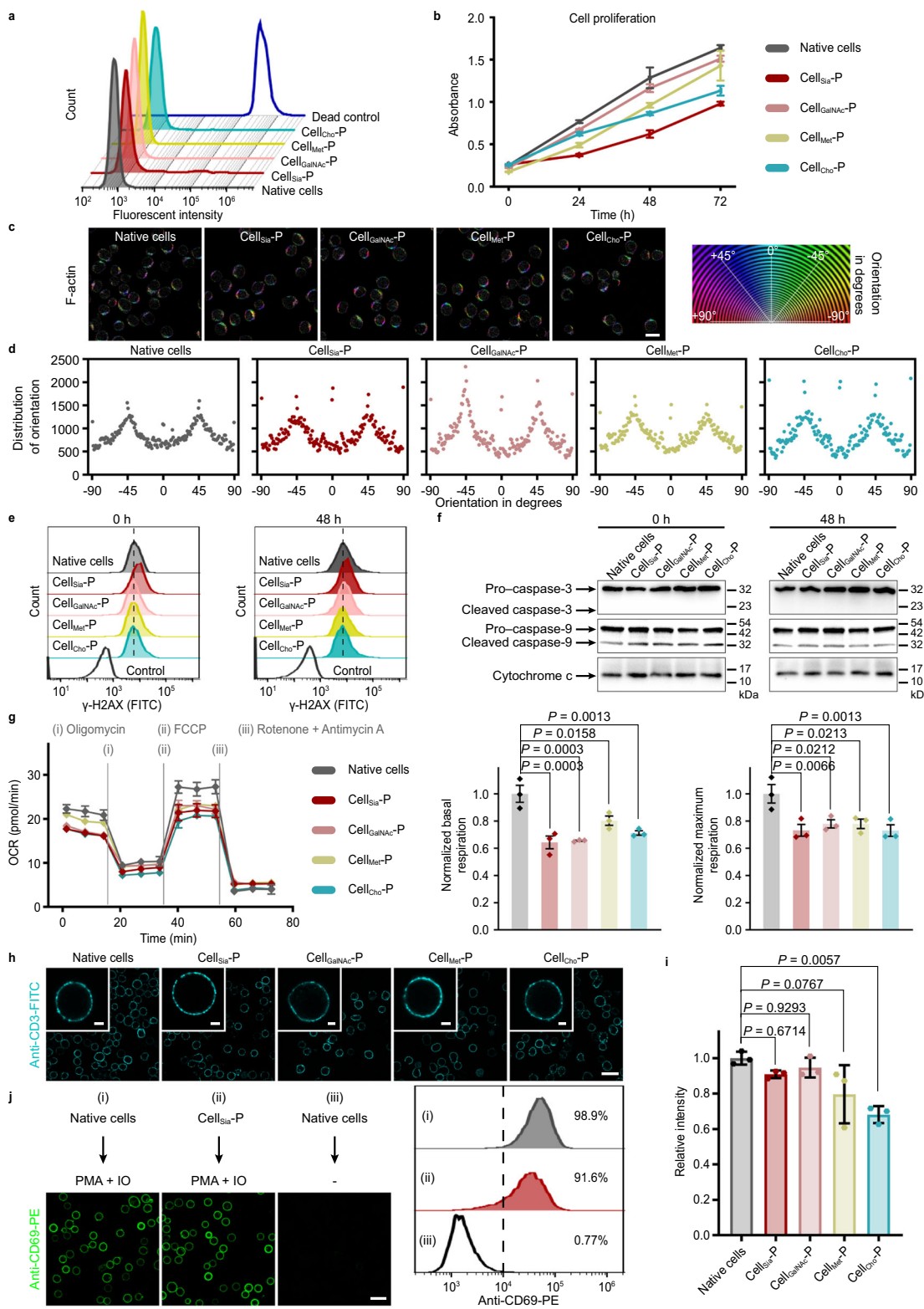

more than 11%) in Cell$_{Sia}$-P and Cell$_{GalNAc}$-P groups. Inhibition was less effective when polymers were grown at protein or lipid sites. This suggests that polymers grown at different sites exhibit different degrees of T-cell protection, and that using glycans as growth sites can more efficiently impede lectin-induced T-cell apoptosis. Together, we suggest that the proposed polymerization platform can achieve differential regulation of cell function by tailoring the growth sites, which provides an innovative mechanism and method for the engineering and functional regulation of immune cells.

## Copolymerization of glycan-modified monomers at the outermost glycan sites

Finally, we attempted to use SSP for copolymerization of biomimetic macromolecules on the cell surface to modulate cell recognition behavior. The cell surface is enriched with a layer of glycoconjugates called glycocalyx, which is the primary mediator of cellular interactions with the outside world. However, current research is limited to the insertion of glycopolymers synthesized in solution into cell membranes for glycobiological studies. The inherent properties of

**Fig. 4 | The biological effects of polymers grown in situ on living Jurkat T cells. a** Cell membrane integrity and cellular activity of different Cell-P were measured by FCM using LIVE/DEAD™ fixable far-red dead cell stain kit. **b** Cell proliferation of native cells and Cell-P measured by CCK-8 kit ($n = 8$ independent samples, mean ± SD). **c, d** Orientation diagrams (**c**, left) and corresponding histograms (**d**) of F-actin in different Cell-P and native cells by phalloidin-iFluor 594 staining. The circular color map coding to indicate the orientation of F-actin is shown (**c**, right). Scale bar, 10 μm. **e** Effect of the SSP process on intracellular DNA. Representative FCM immunostaining signals of phosphorylated histone H2AX (γ-H2AX) in Cell-P immediately after polymerization and after 48 h of culture, respectively. Native cells were used as controls. γ-H2AX is a sensitive marker for DNA damage. **f** Western blotting of proteins associated with ROS and hypoxia damage (caspase-3, caspase-9, and cytochrome c). Cropped blots are shown, and the full scans are supplied in the Source Data file. **g** Effect of the SSP process on cellular metabolism. Oxygen

consumption rates (OCRs) of Cell-P were measured using the Seahorse XF Cell Mito Stress Test, and statistically normalized to basal and maximal respiration levels ($n = 3$ independent samples, mean ± SEM). FCCP, carbonyl cyanide-4 (tri-fluoromethoxy) phenylhydrazone. **h, i** Immunostaining images (**h**) and quantitative FCM data (**i**) of Cell-P and native cells undergoing anti-CD3-FITC incubation. Scale bar, 25 μm. Enlarged images show a single stained cell; Scale bar, 3 μm. Data are presented as the mean ± SD of $n = 3$ individual experiments. **j** Immunostaining images and FCM histograms of CD69 expressed on Cell$_{Sia}$-P after stimulation with phorbol 12-myristate 13-acetate (PMA) and ionomycin (IO). Native cells with and without stimulation were used as controls. Scale bar, 25 μm. In **g** and **i** statistical differences were determined by one-way ANOVA with Tukey's multiple comparison test. $P < 0.05$ was considered statistically significant. In **a**–**h** and **j** data are representative of three independent experiments with similar results. Source data are provided as a Source Data file.

polymers such as large molecular weight and chain flexibility lead to low labeling efficiency and long labeling time[63]. Although in situ initiation of polymerization of glycan monomers on cell membranes could in principle overcome these drawbacks, no successful method has been reported due to the difficulty of growing polymers in situ on the cell surface. Therefore, we exploited the possibility of in situ polymerization of glycoconjugates at the outermost glycan sites on the cell surface by taking advantage of the fact that SSP can grow polymers at selected sites.

We first synthesized methacrylamide-mannose (MA-Man) monomer, and in situ copolymerized HPMA and MA-Man at the terminal glycan, Sia, of cellular glycocalyx to obtain poly (HPMA-co-MA-Man)-engineered Jurkat T cells (Cell$_{Sia}$-P$^{Man}$) (Fig. 7a). We used a mannose-specific lectin, Concanavalin A (Con A, FITC modified) to reflect the level of Man on cell surface. Con A has a homotetrameric structure that can readily enter cells as well as aggregate cells by multivalently binding to Man on the cell surface[64]. After incubating Con A-FITC with native cells, Cell$_{Sia}$-P, or Cell$_{Sia}$-P$^{Man}$, there was a strong FITC fluorescence signal at the native cell surface, which entered the cell after 15 min, and the cell aggregation was obvious (Fig. 7b, right). As expected, the FITC fluorescence of the Cell$_{Sia}$-P group was reduced by 65.5% compared to the native group (Fig. 7c, d), and the cells were evenly dispersed without aggregation (Fig. 7b, middle). In contrast, the fluorescence intensity on Cell$_{Sia}$-P$^{Man}$ was significantly higher than that of Cell$_{Sia}$-P (Fig. 7b, left, c, and d), demonstrating the successful growth of Man-containing polymers. We found interesting highlights: (1) unlike native cells, there was no obvious fluorescence signal inside Cell$_{Sia}$-P$^{Man}$ (Fig. 7b, left), indicating that P$^{Man}$ slowed down the entry of Con A into cells; (2) Cell$_{Sia}$-P$^{Man}$ aggregation was significantly weaker than native cells (Fig. 7b, left). The above results suggest that P$^{Man}$, as a biomimetic glycan chain with a different form of mannose distribution, not only exhibits the same function of recognizing Con A as native glycan chains, but also has the function that native glycan chains do not have, that is, weakening the ability of Con A to enter and aggregate cells. This reflects the significance of constructing artificial glycocalyx on living cells and provides a potential pathway for regulating cell communication and signal transduction based on glycan recognition.

We envisioned that the biomimetic glycocalyx constructed on the cell surface by SSP copolymerization strategy can be further remodeled just as native glycans. To demonstrate this, we copolymerized methacrylamide-galactose (MA-Gal) and HPMA on living Jurkat T cells using Sia as the growth site to generate Cell$_{Sia}$-P$^{Gal}$. As seen in Fig. 7e–g and supplementary Fig. 33, the fluorescence signal of Cell$_{Sia}$-P$^{Gal}$ is significantly stronger than that of native cells (2-fold increase). The increase should be attributed to the Gal groups introduced during the copolymerization process. The above two examples illustrate that SSP copolymerization is a general method for introducing defined glycan structures on the cell surface. The

growth of the glycopolymer can be viewed as the process to mimic the growth of glycocalyx, and the synthesized polymer can be used as an artificial glycocalyx to regulate the recognition behavior of the cell.

We have developed a cytocompatible Fenton-RAFT polymerization technique and further established an in situ polymerization platform at selected sites of living cells enabled by metabolic labeling. The Jurkat T cells after polymerization maintained excellent cell viability with essentially unaltered cytoskeleton and immune activability, and intracellular ROS, DNA, and protein levels, as well as cellular metabolism and proliferative capacity, were not significantly affected. The selection of polymer growth site (glycans, proteins, lipids) has a differential effect on the retention time of the polymer on the cell surface and on the multivalent binding of cellular glycans to lectins. In particular, polymers grown from sugar sites can effectively inhibit lectin-induced T-cell apoptosis, which provides an innovative way for glycan axis-based immunomodulation. We have used the proposed platform to synthesize multiple copolymers on living cells, enabling the in situ polymerization of glycomonomers on the surface of mammalian cells, which to the best of our knowledge was never reported before. These biomimetic glycocalyx has distinct recognition properties that differ from natural glycocalyx and can be further edited, thus providing a promising approach to cellular glycocalyx remodeling. Our work suggests fresh ideas for the development of polymer-based cell engineering technology, opens further dimensions for the regulation of cellular functions and behaviors, and holds great potential for cell and tissue engineering, as well as cell-based disease therapy.

## Methods
### Metabolic labeling
Glycan labeling: Jurkat T cells ($2 \times 10^5$ cells/ml) were allowed to incubate with 40 μM of tetraacylated N-azidoacetylmannosamine (Ac$_4$ManNAz) or tetraacylated N-azidoacetylgalactosamine (Ac$_4$GalNAz) in RPMI-1640 complete medium for 48 h in a humidified incubator at 37 °C under 5% CO$_2$ to achieve the labeling of cell surface Sia or mucin-type O-GalNAc with azide.

Protein labeling: Jurkat T cells ($5 \times 10^5$ cells/ml) were cultured in serum-free, methionine-free RPMI-1640 medium in a humidified incubator at 37 °C under 5% CO$_2$ for 1 h to deplete residual methionine (Met). Subsequently, the cells were collected by centrifugation and dispersed in methionine-free RPMI-1640 complete medium containing 0.1 mM azidohomoalanine (AHA) and placed in the incubator for 4 h to install AHA to the Met sites of cell surface proteins.

Lipid labeling: Jurkat T cells ($2 \times 10^5$ cells/ml) were allowed to incubate with 1 mM 1-azidoethyl-choline (AECho) in RPMI-1640 complete medium for 24 h in the humidified incubator at 37 °C under 5% CO$_2$ to install azide on the choline (Cho) sites on cell surface.

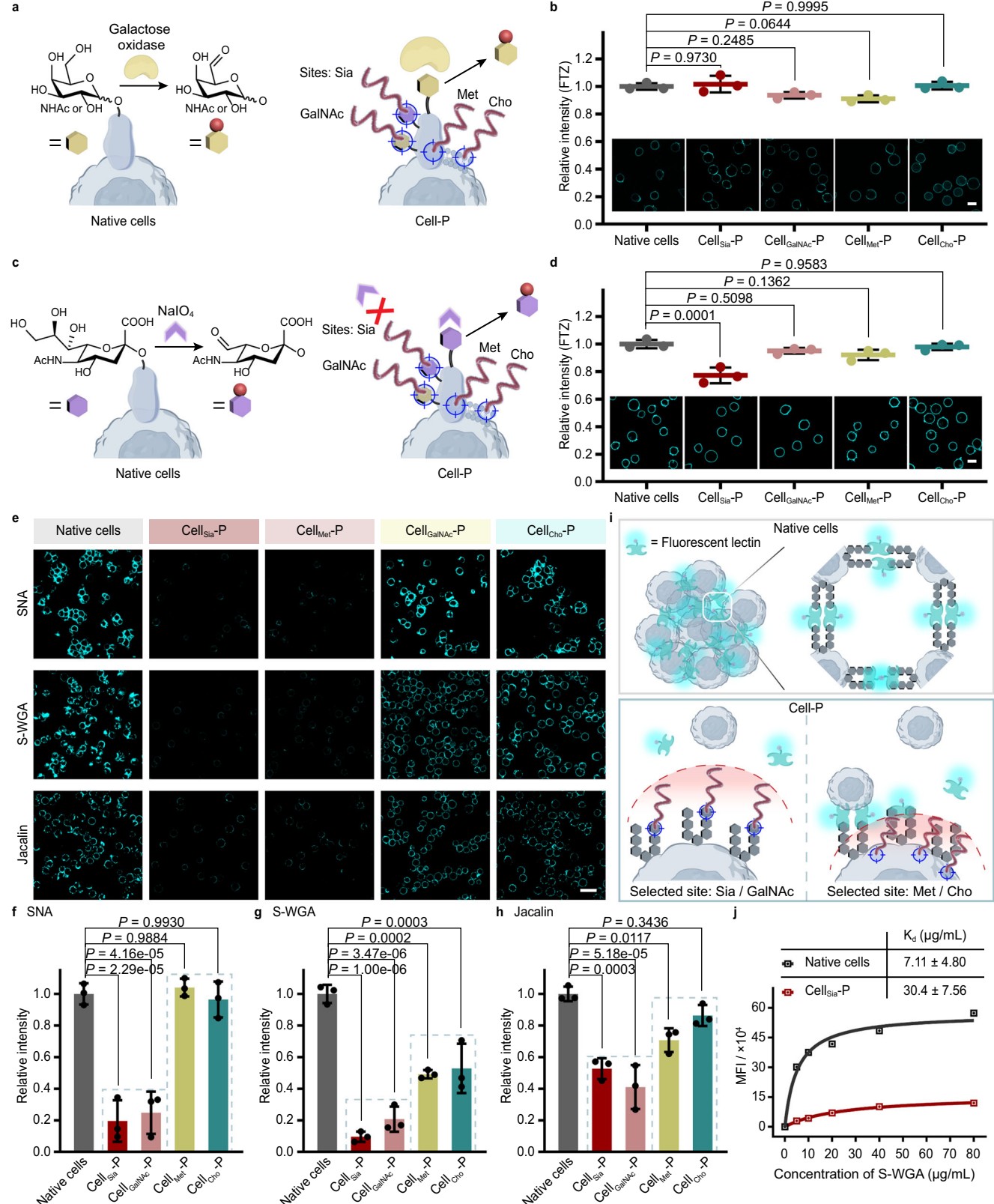

### General protocol of site-selected in situ polymerization (SSP)

(1) Native cells are treated with metabolic reagents to install azide at selected biological sites.

(2) The obtained cells, Cell-N$_3$, were collected by centrifugation and washed three times with phenol red-free RPMI-1640. The cells were evenly dispersed at a density of $1 \times 10^6$ cells/ml in PBS

solution containing $40\,\mu$M DBCO-BTPA[65]. After 1-h reaction at $4\,°$C, the cells were centrifuged and washed three times with phenol red-free RPMI-1640 to obtain cells with CTA installed at selected sites (Cell-CTA).

(3) The following procedure describes typical conditions for in situ polymerization at Cell-CTA. The concentration ratio of [HPMA]:[CTA]:[H$_2$O$_2$]:[Fe$^{2+}$]=80:1:0.3:0.1 was used for the

**Fig. 5 | Differential effects of polymer growth sites on the glycan recognition behavior of cells. a** Schematic depicting the oxidation of -OH at the C-6 position of terminal Gal/GalNAc by galactose oxidase (GAO) to yield an aldehyde group on the surface of native cells and Cell-P with polymer grown from different sites. **b** Quantitative FCM analysis and representative images of native cells and Cell-P after GAO oxidation and fluorescein-5-thiosemicarbazide (FTZ) staining (to indicate generated aldehyde groups). Scale bar, 10 μm. **c** Schematic depicting the oxidation of terminal Sia by the small molecule $NaIO_4$ to generate an aldehyde group on the surface of native cells and Cell-P with polymer grown from different sites. **d** Quantitative FCM analysis and representative images of native cells and Cell-P after $NaIO_4$ oxidation and FTZ staining. Scale bar, 10 μm. **e–h** Effect of polymer growth sites on lectin binding to cell surface glycans. CLSM images (**e**) and quantitative FCM data (**f–h**) of native cells and Cell-P after incubation with FITC-conjugated Sambucus Nigra lectin (SNA, **f**), succinylated wheat germ agglutinin (S-WGA, **g**), or Jacalin (**h**). Recognition preference: SNA, α2,6-Sia; S-WGA, N-acetylglucosamine; Jacalin, galactosyl (β–1,3) N-acetylgalactosamine. Scale bar, 25 μm. **i** Proposed binding mechanism of lectins towards native cells and different Cell-P. **j** Saturation binding experiments of S-WGA-FITC to native cells and $Cell_{Sia}$-P. Binding isotherms were fitted with one-site binding model to obtain the binding affinities ($K_d$). MFI, mean fluorescence intensity. CLSM images and **j** are representative of three independent experiments with similar results. In **b**, **d**, **f–h**, data are shown as the mean ± SD of $n = 3$ individual experiments. Statistical differences were determined by one-way ANOVA with Tukey's multiple comparison test. $P < 0.05$ was considered statistically significant. Source data are provided as a Source Data file.

preparation of PolyHPMA homopolymer-engineered cells (Cell-P). (Preparation of functionalized polymer-engineered cells at selected sites, as described infra.) The polymerization medium was prepared by adding 100 μl of B27 supplement minus AO to 4900 μl of phenol red-free RPMI-1640, followed by adjusting the pH to 6.6 - 6.9. The polymerization was performed in 1.5 ml sterile glass vials equipped with rubber septum. 120 μmol HPMA, 1.5 μmol BTPA and 0.45 μmol $H_2O_2$ dissolved in 190 μl of polymerization medium were added to the glass vial and sealed. The mixture was bubbled with nitrogen for 30 min. Cell-CTA ($2 \times 10^6$ cells), obtained by Step (2), were dispersed in 100 μl of pre-deoxidized polymerization medium. The cell suspension was rapidly added to the glass vial with the cap open and sealed immediately. The inlet needle connected to nitrogen was inserted through the rubber septum into the space above the purge level, and exhaust needle was passed through the septum to expel the gas and circulate the nitrogen to remove the residual air from the upper layer, and all needles were withdrawn after 3 min. The polymerization was initiated by adding 10 μl of ammonium ferrous sulfate solution (15 mM) through a degassing syringe. The reaction was carried out in a shaker at 200–250 rpm for 2 min (or 1 min), then terminated by exposing the reaction solution to air and immediately adding 700 μl of RPMI-1640 complete medium to dilute the reaction solution. The cells after polymerization were collected by centrifugation at 4 °C, and the pellet was washed three times with phenol red-free RPMI-1640, and resuspended in 1 ml of phenol red-free RPMI-1640 to obtain Cell-P. The supernatant was added with $D_2O$ for monomer conversion analysis using $^1H$ NMR.

(4) For chain extension, Cell-P was subjected to a new round of in situ polymerization as described in Step (3) to yield Cell-PP.

(5) In order to characterize the polymer grown on cell surface, a cleavable CTA, DBCO-SS-BTPA, was used to replace DBCO-BTPA to perform the same procedure as described in Steps (2) and (3). Then the cells were dispersed in PBS solution containing 50 mM tris(2-carboxyethyl)phosphine hydrochloride (TCEP) and incubated in the dark for 30 min. The supernatant was collected by centrifugation and dialyzed in water using MWCO = 1 K dialysis tubing for 48 h. The cleaved polymer was lyophilized prior to GPC measurement.

(6) For the adherent cell line MCF-7, Steps (1) and (2) were performed with cells in adherent state. Then, the cells were treated with trypsin at 37 °C for 2 min, collected by centrifugation, and well dispersed in pre-deoxidized polymerization medium prior to Step (3).

## Copolymerization of functional monomer

(1) Jurkat T cells with CTA installed at Sia sites ($Cell_{Sia}$-CTA) were subjected to SSP-based copolymerization to prepare poly (HPMA-co-AA-$PEG_4$-biotin)-engineered Jurkat T cells ($Cell_{Sia}$-$P^{biotin}$). The concentration ratio for copolymerization is [HPMA + AA-$PEG_4$-biotin]:[BTPA]:[$H_2O_2$]:[$Fe^{2+}$] = (70 + 10): 1: 0.3: 0.1 (350 mM HPMA, 50 mM AA-$PEG_4$-biotin, 5 mM BTPA, 1.5 mM $H_2O_2$, 0.5 mM $Fe^{2+}$), and the operation was the same as that described in Step (3) of General protocol of site-selected in situ polymerization (SSP). Cells labeled only with $Ac_4ManNAz$ (yielding $Cell_{Sia}$-azide) were also subjected to the copolymerization process.

(2) Jurkat T cells with CTA installed at GalNAc, Met or Cho sites (Cell-CTA) were subjected to SSP-based copolymerization to prepare poly (HPMA-co-AA-$PEG_4$-azide)-engineered Jurkat T cells (Cell-$P^{azide}$). The concentration ratio for copolymerization is [HPMA + AA-$PEG_4$-azide]:[BTPA]:[$H_2O_2$]:[$Fe^{2+}$] = (70 + 10): 1: 0.3: 0.1 (350 mM HPMA, 50 mM AA-$PEG_4$-azide, 5 mM BTPA, 1.5 mM $H_2O_2$, 0.5 mM $Fe^{2+}$), and the operation was the same as that described in Step (3) of General protocol of site-selected in situ polymerization (SSP).

## Visualization of copolymers on living cell surface

(1) For $Cell_{Sia}$-$P^{biotin}$: Four cell samples, $Cell_{Sia}$-$P^{biotin}$, $Cell_{Sia}$-azide undergoing copolymerization, $Cell_{Sia}$-CTA, and $Cell_{Sia}$-azide, were prepared as described above. For each sample, cells were dispersed at $1 \times 10^6$ cells/ml in 20 μg/ml SA-Cy3-containing PBS and incubated for 1 h at 4 °C in the dark. The cells were washed three times with 1% FBS-containing PBS by centrifugation, and dispersed in 1 ml of PBS. The SA-Cy3 fluorescence signal was analyzed using a CytoFLEX flow cytometer equipped with a 488 nm laser and a 585/42 nm filter and CLSM imaging (ex/em: 554/565-615 nm).

(2) For Cell-$P^{azide}$: Cell-$P^{azide}$, Cell-azide, Cell-CTA, and native Jurkat T cells were each dispersed at $1 \times 10^6$ cells/ml in 10 μM DBCO-Cy5-containing PBS and incubated for 30 min at 4 °C in the dark. After three washes with 1% FBS-containing PBS by centrifugation, the cells were evenly dispersed in PBS, and immediately analyzed with a CytoFLEX flow cytometer equipped with a 638 nm laser and a 660/10 nm filter, and imaging with CLSM (ex/em: 646/655–705 nm). For STED imaging, 20 μM Click-iT™ sDIBO was used to replace DBCO-Cy5, with other conditions unchanged (ex/em: 488/500–550 nm).

## Cleavage of Sia by NEU

$Cell_{Sia}$-$P^{biotin}$ ($1 \times 10^6$ cells) were incubated in PBS (pH = 5.8) containing 0.2 U/ml NEU at 37 °C for 1 h. After three washes with 1% FBS-containing PBS, the cells were stained with SA-Cy3 and analyzed with a CytoFLEX flow cytometer. Untreated $Cell_{Sia}$-$P^{biotin}$ and native cells were used as controls.

## T-cell stimulation

$1 \times 10^6$ cells/ml of $Cell_{Sia}$-P (or native Jurkat T cells) were treated with 10 ng/ml phorbol 12-myristate 13-acetate (PMA) and 1 μg/ml ionomycin (IO) in RPMI-1640 complete medium for 4 h in a humidified incubator at 37 °C under 5% $CO_2$. The cells were then centrifuged and washed

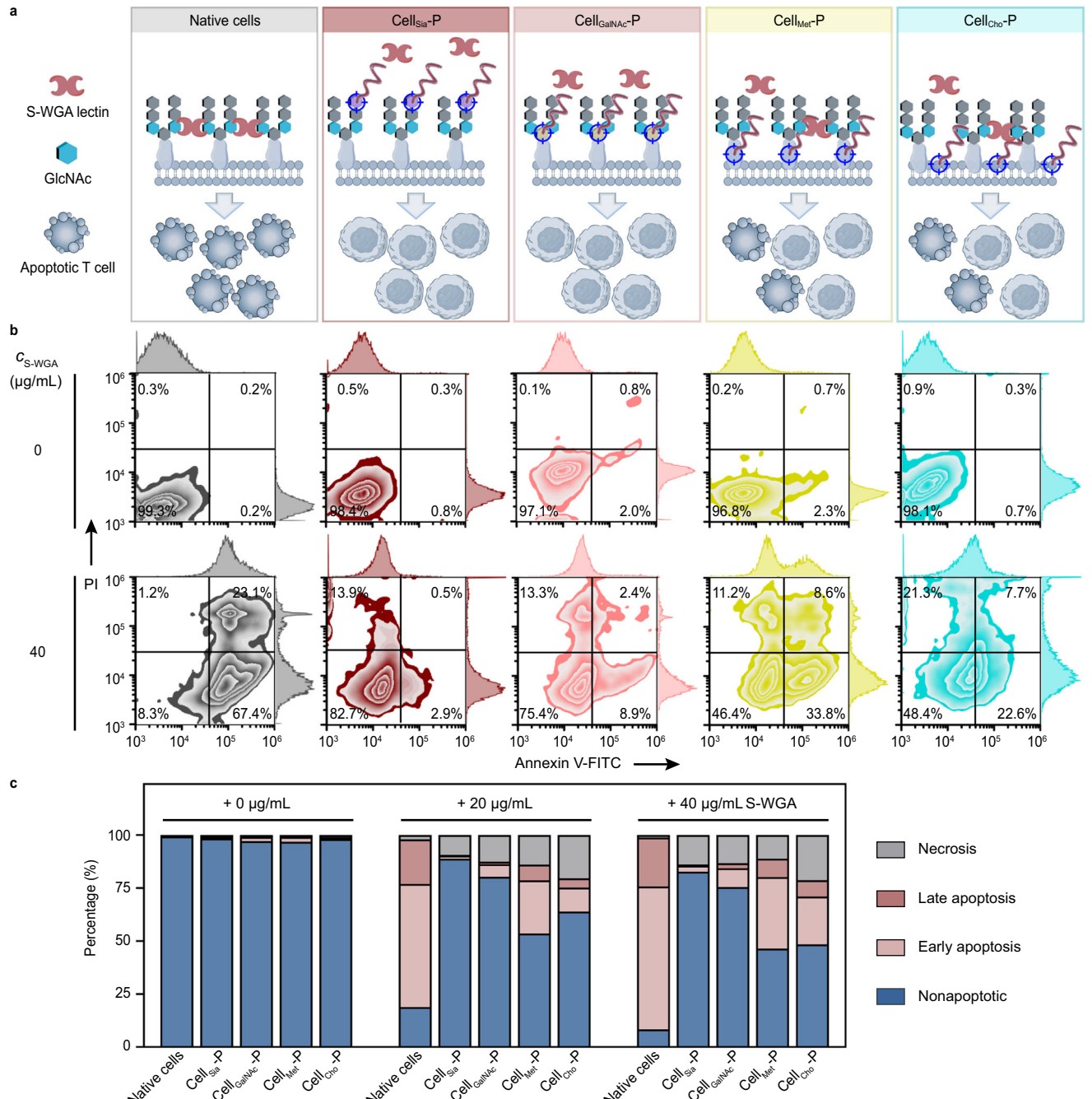

**Fig. 6 | Polymers grown at different sites inhibited lectin-mediated T-cell apoptosis to varying degrees. a** Schematic illustration of the differential distribution of S-WGA with multivalent recognition capability towards N-acetylglucosamine (GlcNAc) after incubation with native cells and different Cell-P, which led to different degrees of T-cell apoptosis. **b, c** Zebra plots (**b**) and

the corresponding data (**c**) for cell apoptosis analysis after S-WGA treatment using Annexin V-FITC/propidium iodide (PI) staining. $c_{S-WGA}$, concentration of S-WGA. Polymers grown from glycan sites could substantially alleviate S-WGA mediated T-cell apoptosis. Data are representative of three independent experiments with similar results.

three times with PBS containing 1% Fetal bovine serum (FBS) for immunofluorescence staining of cell surface activation marker CD69. Native Jurkat T cells without PMA + IO treatment were used as controls.

**Immunofluorescence staining**

For immunofluorescence staining of CD69, cells were resuspended at $1 \times 10^6$ cells/ml in PBS containing PE anti-human CD69 antibody (1:20 dilution, BioLegend #310905) and incubated for 30 min at 4 °C in the dark. After three washes with 1% FBS-containing PBS to remove excess unreacted antibody, the samples were resuspended in PBS buffer and immediately analyzed using CytoFLEX flow cytometry equipped with a

488 nm laser and a 585/42 nm filter and imaged using CLSM (ex/em: 540/550–600 nm).

For immunofluorescence staining of CD3, cells at $1 \times 10^6$ cells/ml were fixed with 4% paraformaldehyde (PFA) for 15 min at r.t., and washed three times with 1% FBS-containing PBS by centrifugation. Cells were then incubated with PBS containing FITC anti-human CD3 antibody (1:20 dilution, BioLegend #300305) for 30 min at 4 °C in the dark, washed three times with 1% FBS-containing PBS, and resuspended in PBS. The samples were analyzed by CytoFLEX flow cytometry equipped with a 488 nm laser and a 525/40 nm filter, and imaged by CLSM (ex/em: 495/500–550 nm).

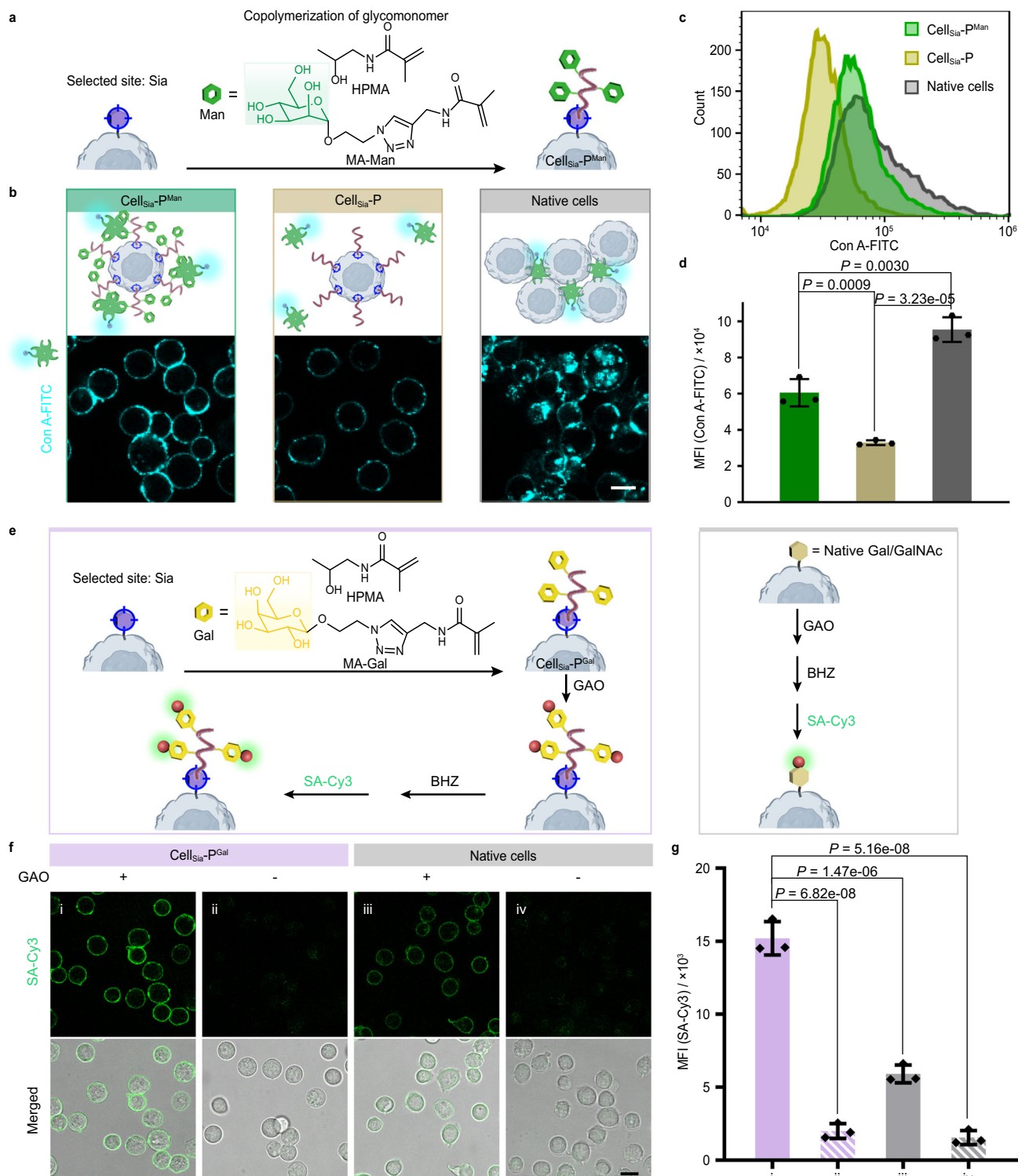

**Fig. 7 | In situ copolymerization of glycomonomers at the outermost glycan sites. a** Schematic showing the preparation of $Cell_{Sia}\text{-}P^{Man}$ by in situ copolymerization of HPMA and MA-Man at Sia site. **b**–**d** Multivalent recognition of Man on $Cell_{Sia}\text{-}P^{Man}$, $Cell_{Sia}\text{-}P$, and native cells by Concanavalin A (Con A)-FITC. **b** CLSM images, scale bar, 10 μm. **c**, **d** FCM histograms and data. **e** $Cell_{Sia}\text{-}P^{Gal}$ was prepared by in situ copolymerization of HPMA and MA-Gal at Sia site. The introduced Gal can be remodeled by GAO, generating aldehyde groups that can be detected by biotin hydrazide (BHZ) labeling and SA-Cy3 staining (left, purple). Natural terminal Gal/ GalNAc on cell surface can also be oxidized by GAO (right, gray). **f**, **g** CLSM imaging (**f**) and FCM analysis (**g**) of $Cell_{Sia}\text{-}P^{Gal}$ and native cells with and without treatment by GAO, followed by BHZ and SA-Cy3 labeling. Scale bar, 10 μm. In **b**, **c**, and **f**, data are representative of three independent experiments with similar results. In **d**, **g**, data are shown as the mean ± SD of $n = 3$ individual experiments. Statistical differences were determined by one-way ANOVA with Tukey's multiple comparison test. $P < 0.05$ was considered statistically significant. Source data are provided as a Source Data file.

For F-actin staining, cells at $1 \times 10^6$ cells/ml were fixed with 4% PFA for 15 min at r.t. After centrifuging and gently aspirating the fixing solution, the fixed cells were washed twice in PBS containing 1% FBS. PBS containing 0.2% Triton X-100 was added to treat the fixed cell pellet for 5 min to increase permeability. The permeabilized cells were centrifuged and washed twice in 1% FBS-containing PBS. Cells were incubated in 1% FBS-containing PBS supplemented with phalloidin-iFluor 594 (1:1000 dilution, Abcam #ab176757) for 30 min at r.t. Cells were collected by centrifugation, washed three times in 1% FBS-containing PBS. The cells were stained with 4′,6-diamidino-2-phenylindole dihydrochloride (DAPI) for 5 min, washed three times in 1% FBS-containing PBS, and resuspended in PBS. Stained cells were analyzed using a CytoFLEX flow cytometer equipped with a 488 nm laser and a 585/42 nm filter. The stained cell samples were also imaged using CLSM. For phalloidin-iFluor 594, ex/em: 590/600–650 nm; for DAPI, ex/em: 405/425–475 nm. The cell images with F-actin stained were analyzed for orientation using the OrientationJ plug-in of ImageJ software.

For γ-H2AX staining, cells at $1 \times 10^6$ cells/ml were fixed and permeabilized as described above. Cells were then incubated in 1% FBS-containing PBS supplemented with monoclonal antibody (mAb) specific for phosphorylated histone H2AX (γ-H2AX) (1:200 dilution, Cell Signaling Technology #9718) for 60 min at r.t. After centrifugation and five washes with 1% FBS-containing PBS, cells were incubated in 1% FBS-containing PBS supplemented with secondary antibody goat anti-rabbit IgG H&L (Alexa Fluor 488, pre-adsorbed, 1:1000 dilution, Abcam #ab150081) for 30 min at r.t. Cells were collected by centrifugation, washed three times in 1% FBS-containing PBS, and resuspended in PBS. Stained cells were analyzed using a CytoFLEX flow cytometer equipped with a 488 nm laser and a 585/42 nm filter. Cell samples were also imaged using CLSM.

### Western blotting
Cells were dispersed in precooled RIPA lysis buffer containing protease inhibitor and phenylmethylsulfonyl fluoride and lysed on ice for 30 min, with vortexing every 10 min. Cell lysates were centrifuged at $16,000\,g$ for 20 min at 4 °C and the supernatants were collected for protein quantification using the BCA assay kit.

Proteins were separated by 12% SDS-PAGE (protein loading was 20 µg) and transferred to PVDF membranes. After washing with TBST (Tris-buffered saline containing 0.1% Tween 20), the PVDF membranes were blocked in TBST containing 3% BSA for 60 min at r.t. Antibodies (caspase-3, Cell Signaling Technology #14220; caspase-9, Cell Signaling Technology #9502; cytochrome c antibody, Cell Signaling Technology #4272) were diluted (1:1000) with TBST containing 3% BSA and incubated with PVDF membranes overnight at 4 °C. The membranes were washed three times with TBST for 15 min each time. The membranes were then incubated with goat anti-rabbit IgG conjugated to HRP (1:5000 dilution, Beyotime #A0208) for 60 min at r.t. After three washes (15 min each) with TBST, the membranes were developed with FDbio-Dura ECL reagent and bands were visualized using the Bio-Rad ChemiDoc XRS+ imaging system. Uncropped and unprocessed scans of the blots are provided in Source Data file.

### Cellular metabolism assay
Cell plates were coated with 100 µg/ml poly-D-lysine for 5 min, washed with water, and dried. Cell-azide, Cell-CTA, Cell-P, and native cells were dispersed in assay medium (RPMI-1640 medium containing 1 mM pyruvate, 2 mM glutamine, and 10 mM glucose) and added to the cell plates ($5.0 \times 10^4$ cells/well). 130 µl of assay medium was carefully added to each well and the cell plates were transferred to a $CO_2$-free incubator at 37 °C and incubated for 1 h. The sensor cartridges were prehydrated overnight and calibrated on the day of the assay (at 37 °C for 1 h). Oligomycin (2.5 µM), FCCP (2 µM) and rotenone / antimycin A (0.5 µM)

were injected into the built-in port and measured using the Seahorse XFe96 Analyzer.

### GAO oxidation and fluorescent labeling of cell surface terminal Gal/GalNAc (galactose/N-acetylgalactosamine)
Cell$_{Sia}$-P, Cell$_{GalNAc}$-P, Cell$_{Met}$-P, Cell$_{Cho}$-P, and native cells were each suspended in 0.1 mg/ml GAO-containing PBS ($1 \times 10^6$ cells/ml) and incubated at r.t for 30 min. Cells were collected by centrifugation at 4 °C and washed three times with 10 mM galactose-containing PBS. Then, the cell pellet ($1 \times 10^6$ cells) was well dispersed in 1 ml of PBS containing 5% FBS, 10 mM aniline and 100 µM fluorescein-5-thiosemicarbazide (FTZ), and incubated at 4 °C for 30 min. Then cells were washed three times with 1% FBS-containing PBS, pelleted again, and resuspended in PBS buffer. Flow cytometry analysis using a 488 nm laser and 525/40 nm filter was performed immediately. These cell samples were also imaged by CLSM (ex/em: 495/500–550 nm).

### NaIO$_4$ oxidation and fluorescent labeling of cell surface terminal Sia (Sialic acids)
$1 \times 10^6$ cells were dispersed in 1 ml precooled PBS solution containing 1 mM NaIO$_4$, and incubated at 4 °C for 15 min. The cells were then pelleted by centrifugation at 4 °C, followed by aspiration of the supernatant and three washes with PBS. The cells were then stained with FTZ and analyzed, as described for the GAO oxidation experiment.

### Lectin recognition of cell surface glycans
Cells of $1 \times 10^6$ cells/ml were incubated with 10 µg/ml Sambucus nigra lectin (SNA)-FITC (Vector Labs #FL-1301-2); succinylated wheat germ agglutinin (S-WGA)-FITC (Vector Labs #FL-1021S-5); Jacalin-FITC (Vector Labs #FL-1151-5), respectively, for 30 min at 4 °C in the dark. The incubation buffer for these three lectins consisted of 10 mM HEPES, 0.15 M NaCl, and 0.1 mM CaCl$_2$ (pH 7.5). The cells were then washed at least three times with 1% FBS-containing PBS by centrifugation and resuspended in precooled PBS.

### Monitoring of lectin-induced T-cell apoptosis
Cells were incubated at $1 \times 10^6$ cells/ml in HEPES buffer (consisting of 10 mM HEPES, 0.15 M NaCl, and 0.1 mM CaCl$_2$; pH 7.5) containing 0, 20 or 40 µg/ml S-WGA (Vector Labs #L-1020) for 4 h at 4 °C. Cells were then washed three times with 1% FBS-containing HEPES buffer by centrifugation. $5 \times 10^5$ cells were dispersed in 500 µl of binding buffer of annexin V-FITC/propidium iodide (PI) apoptosis assay kit, followed by the addition of 5 µl of Annexin V-FITC and 5 µl of PI. After incubation at r.t. for 10 min in the dark, the cells were analyzed by CytoFLEX flow cytometry (488 nm laser, 525/40 nm (Annexin V-FITC) and 585/42 nm (PI) filters).

### Copolymerization of glycomonomers at Cell$_{Sia}$-CTA
Poly (HPMA-co-MA-Man)-engineered Jurkat T cells with Sia as the growth site (Cell$_{Sia}$-P$^{Man}$) was prepared using Cell$_{Sia}$-CTA by adjusting the ratio of polymerization components in the SSP process to [HPMA + MA-Man]:[BTPA]:[H$_2$O$_2$]:[Fe$^{2+}$] = (72 + 8): 1: 0.3: 0.1 (360 mM HPMA, 40 mM MA-Man, 5 mM BTPA, 1.5 mM H$_2$O$_2$, 0.5 mM Fe$^{2+}$). Cell$_{Sia}$-P$^{Man}$, Cell$_{Sia}$-P, and native Jurkat T cells ($1 \times 10^6$ cells/ml) were each incubated with HEPES buffer (consisting 10 mM HEPES, 0.15 M NaCl, 0.1 mM CaCl$_2$, and 0.01 mM MnCl$_2$; pH 7.5) containing 5 µg/ml Concanavalin A (Con A)-FITC (Vector Labs # FL-1001-25) at 4 °C for 15 min in the dark. Cells were washed three times with 1% FBS-containing PBS by centrifugation at 4 °C, and dispersed in precooled PBS buffer. Cells were immediately analyzed using a CytoFLEX flow cytometer equipped with a 488 nm laser and 525/40 nm filters, and imaged using CLSM (ex/em: 495/500–550 nm).

Poly (HPMA-co-MA-Gal)-engineered Jurkat T cells with Sia as the growth site (Cell$_{Sia}$-P$^{Gal}$) was prepared using Cell$_{Sia}$-CTA by adjusting the ratio of polymerization components in the SSP process to

[HPMA + MA·Gal]:[BTPA]:[H$_2$O$_2$]:[Fe$^{2+}$] = (72 + 8): 1: 0.3: 0.1 (360 mM HPMA, 40 mM MA-Gal, 5 mM BTPA, 1.5 mM H$_2$O$_2$, 0.5 mM Fe$^{2+}$).

To oxidize cell surface exposed Gal/GalNAc by GAO, Cell$_{Sia}$-P$^{Gal}$ and native Jurkat T cells were each suspended in 0.1 mg/ml GAO-containing PBS at a density of $1 \times 10^6$ cells/ml, and incubated at r.t. for 30 min. The cells were washed three times with 10 mM Gal-containing PBS by centrifugation at 4 °C to remove the adsorbed GAO.

To visualize the GAO oxidation of cell surface Gal/GalNAc, four groups of cell samples were prepared: native Jurkat T cells; Cell$_{Sia}$-P$^{Gal}$; native Jurkat T cells after GAO treatment; Cell$_{Sia}$-P$^{Gal}$ after GAO treatment. Cells from each group were dispersed at a density of $1 \times 10^6$ cells/ml in PBS containing 5% FBS, 10 mM aniline, and 100 μM biotin hydrazide (BHZ), and incubated for 30 min in the dark at 4 °C. Cells were then washed three times with PBS, pelleted again, and incubated with 15 μg/ml SA-Cy3-containing PBS for 30 min at 4 °C. After washing with 1% FBS-containing PBS by centrifugation, the cells were resuspended in PBS, and immediately analyzed by FCM and CLSM.

### Statistical analysis

Statistical analysis and graph plotting were performed with GraphPad Prism 8 software. Statistical differences were determined by one-way ANOVA with Tukey's multiple comparison test or two-tailed unpaired Student's $t$-test. $P < 0.05$ was considered statistically significant. The $K_d$ values were obtained by fitting with one-site binding model to the binding isotherm.

### Reporting summary

Further information on research design is available in the Nature Portfolio Reporting Summary linked to this article.

## Data availability

The data generated in this study are provided in the Main Text, Supplementary Information and Source Data files, and are available from the corresponding authors upon request. Source data are provided with this paper.

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

## Acknowledgements

We gratefully acknowledge support from the National Natural Science Foundation of China (21974067 and 22274073 to LD, 21708019 to XW), Fundamental Research Funds for the Central Universities (020514380309 and 021414380502 to LD, 2022300324 to HJ), and the State Key Laboratory of Analytical Chemistry for Life Science (5431ZZXM2305 and 5431ZZXM2204 to LD).

## Author contributions

Y.Z. and L.D. conceived the idea. Y.Z. designed and performed the polymerization and cell-related experiments. L.J.X. cultured Jurkat and MCF-7 cells and purified HPMA and DBCO-BTPA. L.X. synthesized the glycomonomers and performed characterization. C.Y. and G.W. performed the WB experiments. Y.G. co-analyzed cell-related data. S.C. assisted with cell culture and metabolic labeling. X.T. synthesized AECho. C.W. synthesized Ac₄GalNAz. R.X., X.W., H.J. and L.D. supervised the study. Y.Z. and L.D. wrote the manuscript with the input of all authors.

## Competing interests

The authors declare no competing interests.
