## [Peer Review File · Nature Communications]

Site-Selected in situ Polymerization for Living Cell Surface EngineeringEditorial Note: Parts of this Peer Review File have been redacted as indicated to maintain the confidentiality of unpublished data.

REVIEWER COMMENTS

Reviewer #1 (Remarks to the Author):

This manuscript describes in situ polymerization on floating cell line and examined the protective effect after the coating. The cell surface engineering is an emerging technique and has been paid lots of attention with the progress of cell and organ transplantation. Their approach is promising for realizing cell surface protection with polymer layers, however, the proposed methods and results are not originally invented here, but most of them were already reported by some other groups. Therefore, the reviewer unfortunately could not find the novelty and new findings from this manuscript, which was also written in line 68 by authors (combination of metabolic labelling and click reactions). Specific comments are following.

1. Cell surface modification with polymer has been studied for transplantation some time ago. Authors carefully discussed this issue by citing the following papers (these are examples).

Advanced Materials (Weinheim, Germany) (2020), 32 (13)

Langmuir 2020, 36, 41, 12088–12106

Acta Biomater. 2019, 90, 21– 36,

Nano Today (2011), 6 (3), 309-325

Soft Matter (2010), 6 (6), 1081-1091

Advanced Drug Delivery Reviews (2008), 60 (2), 124-145

Chemical Reviews (Washington, DC, United States) (2018), 118 (4), 1664-1690

2. Current focus in the cell surface engineering is an issue of retention time on the cell surface. It is still difficult to control the time. Although the authors showed some results on the retention time in Figure 5F, this is not enough. In particular modified polymers should be studied and discussed the influence of molecular weight on the stability.

3. Line116: this is a serious problem when this technique will be applied to therapeutic cells. Did they check the PO₂, oxygen concentration and also the influence of ischemic state on cell phenotype? Ischemic state can activate the cells.

4. Line 122: The reviewer thinks that Fenton reaction is not ideal since OH radical can possibly damage DNA and cytoplasmic proteins, which cannot be controlled due to chain reaction of radical. Did they prove that DNA and cytoplasmic proteins are not attacked? Long-term cell cytotoxicity needs to be considered when Fenton reaction is used.

5. Line 128: The medium contains cystine. Any influence? Culture medium may not be necessary if it is only 2 min reaction.

6. Line 133: The monomer conversion seems to be low. When compared with direct polymer modification by lipid insertion or covalent attachment, this is disadvantageous. This needs to be

discussed.

7. Line 137 and 453: The conclusion says “The polymerization reaction is very fast, completed in 2 min” however, this cannot be a conclusion since higher cytotoxicity with longer reaction time. They had to stop the reaction.

8. Line 149-158: This is confusion. Did authors show the results on selective site modification with N3 group on cell surface? They only provided results on flow cytometric analysis which just showed the amount of modification. How did they prove the selective modification?

9. Line 166-168: This sentence is difficult to understand. Authors are asked to revise.

10. Line 185-187: Is it common to study the cytoskeleton of floating cells? If so, authors should prove the reference papers.

11. Line 211-213: This discussion is not correct. They considered that reduction of anti-CD3 antibody binding was due to brush regime and flexibility of polymers. However, the modification density of polymer and molecular weight of polymer can influence on the antibody binding.

12. Line 238, 265-266: is there any possibility that NaIO₄ can selectively oxidized other sugars on cell surface randomly. How do authors prove that this reaction is specific to targeted sugars? There are so many sugars to be oxidized potentially.

Reviewer #2 (Remarks to the Author):

The authors grew polymers from the surface of cell with the aim to modulate their surface chemistry. This has been already done by others, but here the authors used the method developed by Bertozzi and co-workers to place the attachment points of a chain transfer agent to specific positions on the surface. This is interesting, but I think it is also very challenging to prove that a) polymers grow from the surface and b) they are placed in specific positions only. The authors have tried to tackle both points, but I am not sure if they fully succeeded as you can see from my comments. I think it is a very difficult system and it is probably not possible to fully tackle the details.

The authors assume that by observing a decline in monomer concentration that polymerization occurred. This is however not sufficient evidence. Monomer might glue to protein, maybe it has diffused into the cells. Moreover, there is no guarantee that a RAFT process took place. It is known that compounds found in cell growth media can interfere with polymerizations and the authors therefore need to build up their system, supported by molecular weight data via SEC, that this is indeed a controlled process. This might start with polymerization in water using the same concentrations before moving to serum free media as the presence of proteins might not allow SEC analysis.

Is there a way to express the amount of CTA per cells? What happens to this numbers when the cell start proliferating? With each cell division this number should half. Is proliferation affected by labelling, CTA reaction and polymerization? I suppose the processes here need to be carried out quickly before cells change, but it would be useful to learn more about the cell metabolism while the cells are chemically altered

The process here is a grafting from process, which is widely done using RAFT polymerization. However, the researchers usually add free RAFT agent in solution to maintain livingness as during RAFT equilibrium the RAFT agent is lost from the surface and it is unlikely to return if there is no excess RAFT agent. Why was this not done here?

Figure 1C: is it possible that the polymer is simply absorbed on the surface? What does the control look like in the absence of CTA? Figure 5 suggests that the CTA is vital, but I still think it would be useful to think about some controls

I am trying to understand how the polymer was cleaved from the surface. The polymerization was carried out in RPMI media. How exactly were the polymers cleaved and purified in order to obtain such nice NMR spectra. I would have thought that there are plenty of impurities, if not from the cell growth media then from the molecules excreted from the cells

The authors used access of WGA as a means to confirm the chain extension. Is this really a valid approach? All it means is that more polymer is on the surface, but this can also be the case when more polymer is adsorbed. Glycans on the cell surface can form strong H- bonding that can retain some polymer. I think the authors need to add a whole suite of experiments to prove this such as increase in fluorescence intensity as in Figure 5 using appropriate controls. Alternatively this part can be fully removed.

Line 212: why would a polymer have a brush regime? This is usually observed when the grafting density is very high

Figure 2: How does CD3 access change after days of proliferation? The sites should be easier to access as there are less polymers on the surface as cells grow in number while the polymer amount is still the same.

Figure 3 f-h: It is almost possible to group the polymer b and c together as well as d and e. The authors argues that b and d protect the lectin binding sites. Could it also just be that different amounts of polymers are attached to the surface in addition to different anchorage sites? Figure 5 suggest that there is more polymer on the surface with GalNC

Figure 5: This is an important experiment and should have been discussed much earlier. It clearly shows the presence of polymer on the surface and it appears that there is no polymer in the absence of CTA. However, there is no experimental part that describes the experiment in detail such as the amount of functional monomer added. Also were the polymers washed before imaging? Why is there no fluorescent polymer in solution as the Fenton reagents would have started a free radical polymerization.

Figure 5A: labelling with AA-PEF-azide: I am not too sure what these experiments show as the discussion in the text is not clear. The authors use cells with and without CTA, but then did not do any polymerization. There are no fluorescent signals on both cells. What did the authors show here?

Figure 5F: I noticed some uneven distribution of polymer on Cho . Why is that?

Reviewer #3 (Remarks to the Author):

In this manuscript the authors report on their development and investigation of a method to prepare polymeric structures on the membrane of living cells. The manuscript is well written and the data is well presented. I really enjoyed reading it and think that the presented results are certainly of high significance to the field.

However (...there is always a "but"... feeling sorry for being the one raising concerns as I consider the authors' findings to be quite exciting), there are some remaining issues/questions I need to point out, the major ones being the following two:

1) According to the methods section, bioorthogonal azide cyclooctyne ligation on live cells was done at a concentration of 40 μM DBCO-BTPA for 1 hour. Considering the second-order rate of this reaction (known to be $<1 \text{ M}^{-1} \text{ s}^{-1}$) this would result in a click-conversion of only $\sim 10\%$. Hence, most of the metabolically installed azide-tags are not used for further modification. Under these conditions it would actually take >1 day for the reaction to go to completion (99%), and ~ 5 hours to reach 50%. Even more striking, the the reaction as described for "Visualization of copolymers on living cell surface" (10 μM DBCO, 30%) would result in a conversion of only $\sim 1\%$. In both cases it might be even less considering a reaction temperature of 4°C . I'd thus recommend the authors to comment on reaction kinetics and why these specific conditions have been chosen. After all, this affects the construction of polymers on the membrane, but also quantification by fluorescence measurements.

2) The authors have carefully chosen - based on preliminary experiments - to perform polymerization for 2 min. However, I'd recommend to avoid saying that the polymerization is "complete" or "completed in 2 min". More importantly, under these conditions the authors analyzed the on-cell formed polymer and describe a PDI of 1.37 (together with further data in Fig. S6). It seems to me that the authors have conducted all further polymerizations in the same way, which would mean that there is no data on different polymer lengths/sizes. Considering further investigations, in particular when size (or point of attachment) seems to matter (cf. Fig. 3I), I think it's crucial to understand and determine the morphology/size of the polymer corona (at least via modeling). Without such data it's difficult to make claims such as the one related to the data shown in Fig. 3I. Moreover, the images provided in Fig. 5B indicate, as the authors describe it in the main text, the formation of "clusters on the surface of living cells". While this might be related to the described copolymerization of functional monomers, it further raises questions about the morphology of the polymer corona. In addition, it would be interesting to see the results of a few selected experiments when trying different reaction conditions / polymerization

times (thus varying polymer sizes). For instance, what is the effect on lectin binding if Cell(Sia)-P and/or Cell(GalNAc)-P are prepared with reduced polymerization time? Could reduced lectin binding also be a result of structural changes of the glycans rather than steric hindrance? In that sense, please add data for lectin binding of metabolically labeled but not polymerized cells as a control.

Minor comments:

3) Please indicate that CTA installation is done via "strain-promoted azide alkyne cycloaddition (SPAAC)" (in the text and Fig. 1A). Please use "azido-functionalized molecules", "azide tags", "azides", etc. and "click-tags" or "clickable tags" (or similar) rather than "N3" and "anchor sites".

4) I agree that a viability of 87% (Fig. S1) is promising, but it doesn't match the authors' statement "essentially no effect on cellular activity" (actually "viability")

5) Please quantify the data as shown in Fig. S3B. It seems there are at least a few percent of dead cells. Certainly no issue, but providing a number seems more accurate than saying "largely maintained".

6) The authors show that modified cells are still able to proliferate. What happens to the polymer corona when cells divide? How do daughter cells look like / behave? While this would probably require even more additional experiments (very likely too many to reasonably ask for), the authors should at least comment on this and/or discuss.

7) Fig. 3: While it's more or less obvious when looking at the whole figure (in particular 3E), please indicate the cell types a-e in the figure caption. Same for Fig. 2 (e.g. when looking at 2F only without immediately noticing the legend in 2B).

8) Do the authors have any idea about the "density" of metabolically incorporated azide tags on the cell membrane? ...also referring to the "modification amount of CTA" as described by the authors.

9) Sugar monomers have been prepared starting from azido-functionalized protected monosaccharides via copper-catalyzed click chemistry. Is there any specific reason for this strategy, for instance, in contrast to Staudinger reduction of the azide to the respective amines followed by direct attachment of the methacryloyl chloride to avoid the triazole being part of the structure? It might act as a linker/spacer between the sugar moieties and the polymer backbone, but that's hard to say without any comparison.

Response to the reviewers' comments

Reviewer #1

1.0 *This manuscript describes in situ polymerization on floating cell line and examined the protective effect after the coating. The cell surface engineering is an emerging technique and has been paid lots of attention with the progress of cell and organ transplantation. Their approach is promising for realizing cell surface protection with polymer layers, however, the proposed methods and results are not originally invented here, but most of them were already reported by some other groups. Therefore, the reviewer unfortunately could not find the novelty and new findings from this manuscript, which was also written in line 68 by authors (combination of metabolic labelling and click reactions). Specific comments are following.*

Response: We are very grateful to the reviewer for the comments. Your comments have inspired us to reorganize the points of innovation that were not clearly stated in the original manuscript. A clarification of the significance of the manuscript is in order here.

1) Design Concept

We have developed a method to grow polymer *in situ* on the SELECTED sites on living cells. Although metabolic labeling and bioorthogonal reactions are very well-established and widely used techniques, installation of CTA on the cell surface by coupling the two can confer properties never achieved by existing *in situ* polymerization techniques: i) polymer modification of selected biomolecule types and ii) covalent attachment of the polymer to selected sites.

The biophysical and biochemical cues of different types of biomolecules are highly diverse, and polymer modification of a given class of biomolecules on the surface of living cells can facilitate the precise intervention of the biomolecules and enable the artificial polymers to execute their functions on demand. This provides new opportunities to explore the mechanisms of biological processes, construct artificial mimetic systems, and develop new engineered cellular applications.

Covalent modification can address the challenge of low binding stability encountered with lipid insertion and electrostatic adsorption-based methods and facilitate the extension of polymer retention time. At the same time, covalent modification only at the selected sites also avoids unnecessary interference with cells and helps to reduce the cytotoxicity of artificial modifications.

Therefore, we think it is an innovative and ingenious idea to introduce polymer growth sites onto cells using metabolic labeling and bioorthogonal reactions. In the field of protein polymer engineering, site-selective modification has become a mainstream technology (Ref. 5; *Biomacromolecules* **2018**, *19*, 1804-1825) to solve the problem of biotoxicity of non-specific modifications, and we believe that covalent polymer engineering at specific cell sites is also an inevitable development trend, and our work has taken a pioneering step in this regard.

2) Synthesis Method

i) The challenges in growing polymers *in situ* on living cells are: i) cellular viability is not easily maintained; ii) the microenvironment of cell surface is extremely complex and thus polymerization is difficult to occur. To address these two issues, we have explored the first Fenton-RAFT polymerization conditions applicable to living cells due to the advantages of aqueous reaction conditions, a very short induction period, rapid synthesis of controlled polymers at room temperature, and simple apparatus. After polymerization, cell viability, proliferative capacity, cytoskeleton, intracellular ROS levels, DNA and protein levels, cellular metabolic functions, and recognition and activation functions were largely maintained at their original levels.

ii) Based on the developed technology, we have achieved the first *in situ* polymerization of glycoconjugates on living cells. The growth process of the polymer can be regarded as a mimic of glycocalyx development with chemical tools *in situ* on the surface of cells, and the synthesized polymer can be used as an artificial glycocalyx to regulate the recognition behavior of cells.

3) New Findings

i) The effect of the grown polymers on cells is content-dependent and not just a "protection". For example, in cell-protein interactions, polymers grown from the GalNAc site have essentially no effect on anti-CD3 antibody binding but significantly inhibit lectin binding to cells. And this differential effect is related to the nature of the interaction.

ii) Polymers grown at different sites have different cell membrane retention times and different effects on cell recognition behavior, opening new perspectives for the engineering and functional regulation of immune cells.

iv) *In situ* polymerization of glycan monomers at cellular glycocalyx sites yielded biomimetic glycocalyx with properties different from those of the natural glycocalyx, illustrating the importance of constructing artificial glycocalyx on living cells.

We are very grateful to the reviewer for the valuable professional comments. We have considered carefully and supplemented experiments to address each question. These experimental results make the innovative points of the manuscript clearer and more solid, especially in terms of cytocompatibility, site selectivity, and controllability of the method. We have also made the innovation points more explicit in the revised manuscript (lines 71-74; lines 98-99; lines 583-586; lines 594-596).

1.1 Cell surface modification with polymer has been studied for transplantation some time ago. Authors carefully discussed this issue by citing the following papers (these are examples).

Advanced Materials 2020, 32 (13), 1902005.

Langmuir 2020, 36 (41), 12088-12106.

Acta Biomaterialia 2019, 90, 21-36.

Nano Today (2011), 6 (3), 309-325.
Soft Matter (2010), 6 (6), 1081-1091.
Advanced Drug Delivery Reviews (2008), 60 (2), 124-145.
Chemical Reviews (2018), 118 (4), 1664-1690.

Response: Cell surface modification with polymer has been studied for decades and has a wide range of applications in transplantation. Recently, cellular polymer engineering has been expanded for emerging fields such as personalized cell therapy, regenerative medicine, drug delivery, and biosensing/bioimaging. We have carefully read the literature and added the relevant content and references (Refs. 10-12, 14) to the background introduction section (lines 44-46).

1.2 *Current focus in the cell surface engineering is an issue of retention time on the cell surface. It is still difficult to control the time. Although the authors showed some results on the retention time in Figure 5F, this is not enough. In particular modified polymers should be studied and discussed the influence of molecular weight on the stability.*

Response: We are very grateful to the reviewer for the valuable suggestions.

We further manipulated the molecular weight of the polymer (copolymerization of HPMA and AM-PEG₄-azide) by adjusting the polymerization time and continued to culture the obtained Cell-P^{azide} in a normal complete medium for 24 h, followed by observing the retention of polymers grown from different sites (GalNAc, Met and Cho) and with different molecular weights on the cell surface.

In the original manuscript, we showed that monomer conversion increased with increasing polymerization time (Fig. S4D). In revision, we have selected Cell-P^{azide} with 1 and 2 min of polymerization time for comparison. We calculated the retention ratio of polymers (RR-P^{azide}) at 24 h vs. 0 h by DBCO-Cy5 staining of the azide in the polymer and CLSM imaging (Fig. 3F). The retention ratio of the azide tag (RR-azide) for cells with metabolic labeling only (Cell-azide) was also analyzed.

A general pattern was obtained:

- i) For the same site, RR-P^{azide} (2 min) \geq RR-P^{azide} (1 min) > RR-azide;
- ii) For either Cell-P^{azide} or Cell-azide, RR_{GalNAc} > RR_{Met}, RR_{Cho}.

We have added the 1 min polymerization results to the revised manuscript (lines 247-250, 268-270) and updated Fig. 3F (originally Fig. 5F).

1.3 *Line 116: this is a serious problem when this technique will be applied to therapeutic cells. Did they check the PO₂, oxygen concentration and also the influence of ischemic state on cell phenotype? Ischemic state can activate the cells.*

Response: A relationship between ischemic state and cell activation has been reported in the literature (*Nat. Med.* **2009**, *15*, 192-199). Considering that this work deals only with experiments at the cellular level, so we examined the effects of hypoxic

environments on cells.

In the original manuscript, we demonstrated that the 2-min polymerization process had essentially no effect on cell viability (Figs. 4A and S4B), proliferation (Figs. 4B and S4C), and activation (Fig. 4J).

To investigate the effect of the deoxygenation system on the cell phenotype, we added native Jurkat T cells to the polymerization medium that was pre-deoxygenated for 30 min and continued deoxygenation for 2, 5, 10, 30, and 60 min, followed by cell viability (LIVE/DEAD™ Fixable Far-Red Dead Cell Stain Kit), cell proliferation (CKK-8 assay), and apoptosis (Annexin V-FITC/propidium iodide (PI) apoptosis assay kit) assay (Fig. S5). The results showed that the deoxygenation system (2~60 min) did not significantly affect the cell phenotype. The corresponding content has been added to lines 140-141.

We also examined the expression of caspase-3, caspase-9, and cytochrome *c*, which are closely correlated with the cellular hypoxia status, in Cell-P at 0 and 48 h after polymerization by WB experiments. The expression levels of the three proteins were similar to those of native cells (Fig. 4F), indicating that the hypoxic conditions used in the SSP protocol do not affect cytoplasmic proteins. The corresponding content has been added to lines 360-365.

We also examined Cell-P at 0 and 48 h after polymerization by immunofluorescence using CD 69 as a measure of T lymphocyte activation. Using native cells as a control, we confirmed that both types of cell samples were not activated (Fig. S21). The corresponding content has been added to lines 396-399.

The above data indicate that the deoxygenation conditions employed in our study had minimal effect on the cell phenotype.

1.4 Line 122: *The reviewer thinks that Fenton reaction is not ideal since OH radical can possibly damage DNA and cytoplasmic proteins, which cannot be controlled due to chain reaction of radical. Did they prove that DNA and cytoplasmic proteins are not attacked? Long-term cell cytotoxicity needs to be considered when Fenton reaction is used.*

Response: While the OH radical produced by the Fenton reaction may have a negative impact on the cells, the main advantage of this reaction is its aqueous condition, a short induction time and rapid synthesis of a controlled polymer at room temperature. This avoids exposing the cells to organic solvents and the high temperature and pressure environments typically encountered in polymerization reactions. After much investigation, we found that by adjusting the reagent concentration (concentration of initiating radicals) and polymerization medium for the Fenton reaction, we could keep free radical damage under control and ensure that polymerization was completed without affecting cell viability, proliferation, and other phenotypes (Figs. 4 and 6B upper row).

We are grateful to the reviewer for inspiring us to further characterize this. We first

evaluated the reactive oxygen species (ROS) levels in Cell-P (Fig. S16) and found that the ROS levels in Cell_{GalNAc}-P, Cell_{Met}-P, and Cell_{Cho}-P were comparable to those of native cells. There is a slight increase in ROS levels in Cell_{Sia}-P, which may be related to the installation of more initiators at the Sia site (there are 2.9 times as many CTA as on other sites, Fig. S6). The corresponding discussion is added on lines 349-355.

We then performed immunofluorescence staining for the DNA damage marker phosphorylated histone H2AX (γ -H2AX) to examine DNA damage in cells after polymerization (Figs. 4E and S17). Compared with native cells, there was essentially no difference in the fluorescence intensity of Cell_{GalNAc}-P, Cell_{Met}-P, and Cell_{Cho}-P. Again, there was a slight increase in γ -H2AX levels in Cell_{Sia}-P ($\text{MFI}(\text{Cell}_{\text{Sia}}\text{-P}) / \text{MFI}(\text{native cells}) = 1.36$). After further culturing of Cell-P for 48 h, the DNA damage of Cell_{Sia}-P restored and the fluorescence signal was close to the level of native cells ($\text{MFI}(\text{Cell}_{\text{Sia}}\text{-P}) / \text{MFI}(\text{native cells}) = 1.17$). The corresponding discussion is added on lines 356-360.

As mentioned in the response to Comment 1.3, we also examined the expression of the intracellular proteins caspase-3, caspase-9, and cytochrome *c* in Cell-P at 0 and 48 h after polymerization by WB experiments. The expression of these proteins was closely correlated with ROS and hypoxia status. The results showed that the expression levels of these proteins in cells, either immediately after polymerization or after 48-hour culture, were similar to those of native cells (Fig. 4F).

Summarizing the above results, we conclude that our optimized Fenton reaction conditions do not lead to an increase in intracellular ROS levels or the damage of DNA and cytoplasmic proteins under the appropriate CTA installation amount; and the effect of polymerization on cells is further reduced after prolonged culture of Cell-P.

1.5 Line 128: *The medium contains cystine. Any influence? Culture medium may not be necessary if it is only 2 min reaction.*

Response: We performed polymerization experiments in water, HEPES buffer, and culture medium. When the polymerization medium was prepared in HEPES buffer (a common buffer system for cell culture), the polymerization reaction proceeded smoothly, but the cell viability was found to be weakened by calcein-AM fluorescence staining experiment compared to native cells (Fig. S2). In contrast, when the polymerization was performed in the culture medium-based system for 2 min, not only the controlled radical polymerization was successfully initiated, indicating that the amount of cystine in the culture medium did not affect the Fenton-RAFT polymerization, but also the viability and proliferation level of the cells were basically the same as those of native cells (Fig. S4).

1.6 Line 133: *The monomer conversion seems to be low. When compared with direct polymer modification by lipid insertion or covalent attachment, this is disadvantageous. This needs to be discussed.*

Response: Yes, in the work on cellular polymer modification by lipid insertion or covalent linkage, the polymers are synthesized under extracellular, abiotic conditions, which makes it easier to obtain polymers with higher conversion. In contrast, our work addresses the question of how to overcome the spatial limitations of biomolecules and precisely engineer selected sites on cells. This is an area that is not addressed by current approaches. Based on the literature (Ref. 24), we set the monomer conversion to <30%, taking into account that high conversion tends to lead to the loss of end-group functional groups.

1.7 Line 137 and 453: *The conclusion says “The polymerization reaction is very fast, completed in 2 min” however, this cannot be a conclusion since higher cytotoxicity with longer reaction time. They had to stop the reaction.*

Response: We appreciate your point. Indeed, as you said, extending the polymerization time to 10 min does have some effect on cell proliferation (Fig. S4C). In our experiments, 2 min polymerization time was sufficient to achieve the target conversion (<30%, as described in the response to 1.6), so no further extension of polymerization time was necessary. In the revised manuscript, we no longer emphasize the short time required for the method (lines 26, 88, 582-583).

1.8 Line 149-158: *This is confusion. Did authors show the results on selective site modification with N₃ group on cell surface? They only provided results on flow cytometric analysis which just showed the amount of modification. How did they prove the selective modification?*

Response: To grow polymers at the selected sites, we used the "metabolic labeling" technique to incorporate precursors with unnatural moieties (azide) to hijack the specified intracellular synthesis pathway to install CTA at selected sites. We examined four different growth sites: Sia (sialic acid, glycan site) (Ref. 36), mucin-type O-GalNAc (N-acetylgalactosamine, glycan site) (Ref. 37), Met (methionine, protein site) (Ref. 38), and Cho (choline, lipid site) (Ref. 39).

We used the Sia site as a model to demonstrate that polymer modifications do occur at selected sites in the cell membrane.

1) Cells_{Sia}-P^{biotin} was prepared by copolymerizing HPMA and AM-PEG₄-biotin after linking CTA to the Sia site, and the biotin was visualized by SA-Cy3 staining. If the Sia site was not modified with CTA, even if we initiated the polymerization reaction in solution, the monomer in solution and the resulting polymer did not adsorb to the cell surface to give a fluorescent signal (Fig. 2C). The polymer could only be seen on the cell surface when the Sia site was modified with CTA. These data were presented in Fig. 5A in the original manuscript and has been moved to Fig. 2C in the revised manuscript to show that the polymer was indeed grown from the cell surface. The

corresponding description has also been moved and revised (lines 210-230).

2) Neuraminidase (NEU) is the glycosidase that can specifically cleave Sia. In the revised manuscript, we treated Cell_{Sia}-P^{biotin} with NEU and found that the fluorescence binding signal of SA-Cy3 was reduced by 66.5% (Fig. 2D), which was comparable to the typical cleavage ratio of NEU (68.6%, Fig. S10), demonstrating that the polymer grew at the Sia site. The corresponding description has been added to lines 230-235.

The above two experiments demonstrated that the polymer was indeed engineered at the selected biological sites on the cell surface.

1.9 Line 166-168: *This sentence is difficult to understand. Authors are asked to revise.*

Response: We have revised the sentence (lines 175-177).

1.10 Line 185-187: *Is it common to study the cytoskeleton of floating cells? If so, authors should prove the reference papers.*

Response: The actin cytoskeleton is a dynamic filamentous meshwork. It builds the structure of the cell to maintain its basic properties. This physical structure is characterized by continuous remodeling, allowing cells to perform complex motility steps such as directional migration, crossing biological barriers, and interacting with other cells. In particular, the actin cytoskeleton is key to facilitating the passage of T lymphocytes (including the Jurkat T cells used here) through different tissue environments and modulating their stop-and-go behavior as they search for antigen-presenting cells. Because of the importance of actin to T lymphocytes, we investigated the effect of polymer growth on actin expression and orientation.

The relevant literature has been added as Ref. 43 to the revised manuscript.

1.11 Line 211-213: *This discussion is not correct. They considered that reduction of anti-CD3 antibody binding was due to brush regime and flexibility of polymers. However, the modification density of polymer and molecular weight of polymer can influence on the antibody binding.*

Response: We are very grateful to the reviewer for pointing out this error.

To facilitate comparison, in the original manuscript, we adjusted the amount of CTA modified on cellular GalNAc, Met, and Cho sites to be consistent (1.4×10^8 per cell, Fig. S6), while the modification amount of CTA at Sia was 4.0×10^8 per cell, 2.9 times that of the other three conditions (as the model with the highest grafting density).

For the anti-CD3 antibody binding signal, Cell_{Sia}-P and Cell_{GalNAc}-P were similar to native cells, while the signals of Cell_{Met}-P and Cell_{Cho}-P were slightly reduced (by 20.4% and 31.9%, respectively) (Fig. 4I). Therefore, we concluded that, overall, the *in situ* growth of polymers had a limited effect on CD3 accessibility.

For the reduction of signals on Cell_{Met}-P and Cell_{Cho}-P, we speculate that it is because,

the polymer grows closer to CD3 from protein and lipid sites, which may have some effect on CD3 accessibility. We have revised the corresponding descriptions in the revised manuscript (lines 378-379, 381-382).

1.12 *Line 238, 265-266: is there any possibility that NaIO₄ can selectively oxidized other sugars on cell surface randomly. How do authors prove that this reaction is specific to targeted sugars? There are so many sugars to be oxidized potentially.*

Response: If the oxidation conditions (concentration, temperature, time) of NaIO₄ are not controlled, it can oxidize a variety of sugar molecules containing vicinal diol structures. However, a large body of literature (Ref. 55; *Chem. Rev.* **2016**, 116, 14277-14306; *Blood*, **2017**, 129, 3100-3110) has shown that NaIO₄ can selectively oxidize Sia under certain mild conditions.

In the revised manuscript, we used a NEU-based Sia cleavage experiment to demonstrate that the oxidation site of NaIO₄ is indeed Sia (Fig. S23). We oxidized native cells with 1 mM NaIO₄ at 4 °C for 15 min to generate an aldehyde group at the C-7 position of Sia, followed by cleavage of Sia with NEU. The cleavage resulted in a 65.9% reduction in the fluorescence staining signal of the aldehyde group, a proportion consistent with that in the response to Comment 1.8, thus demonstrating that NaIO₄ can specifically oxidize Sia under the experimental conditions chosen.

Reviewer #2

2.1 *The authors grew polymers from the surface of cell with the aim to modulate their surface chemistry. This has been already done by others, but here the authors used the method developed by Bertozzi and co-workers to place the attachment points of a chain transfer agent to specific positions on the surface. This is interesting, but I think it is also very challenging to proof that a) polymers grow from the surface and b) they are placed in specific positions only. The authors have tried to tackle both points, but I am not sure if they fully succeeded as you can see from my comments. I think it is a very difficult system and it is probably not possible to fully tackle the details.*

Response: We are very grateful to the reviewer for finding our work interesting.

For the two points that the reviewer found difficult to prove, we have taken the polymer grown from sialic acid (Sia) sites as a model to demonstrate.

1) Cell_{Sia}-P^{biotin} (CTA+, polymerization+; Fig. 2C, column 1) was prepared by copolymerizing HPMA and AM-PEG₄-biotin after linking CTA to the Sia site, and the polymer side chain biotin was visualized by SA-Cy3 staining. We established 3 control groups: i) CTA-, polymerization+ (column 2); ii) CTA+, polymerization- (column 3); iii) CTA-, polymerization- (column 4). The absence of fluorescence in the third and fourth columns indicates that SA-Cy3 was not adsorbed on the cell surface. If the Sia site was not modified with CTA, even if we initiated the polymerization reaction in solution, the monomer in solution and the resulting polymer did not adsorb to the cell surface to give a fluorescent signal (column 2). The polymer was only visible on the cell surface when the Sia site was modified with CTA (column 1).

These data were included in Fig. 5A in the original manuscript and has been moved to Fig. 2C in the revised manuscript to show that the polymer was indeed grown from the cell surface. The corresponding description has also been moved and revised (lines 210-230).

2) The metabolic labeling technique is a very classical technique for introducing bioorthogonal reporters at selected sites using cellular biosynthetic pathways. Combining this technique with a click reaction ensures that the CTA modification is at the selected site. To demonstrate that the polymer was indeed grown from the Sia site, we performed Sia cleavage experiments using NEU, a glycosidase that can specifically cleave Sia. Treatment of Cell_{Sia}-P^{biotin} with NEU reduced the fluorescence binding signal of SA-Cy3 by 66.5% (Fig. 2D), a ratio comparable to the typical ratio of cell surface sialic acid cleavage using NEU (68.6%, Fig. S10), thus demonstrating that the polymer was indeed grown at the Sia site. The corresponding description has been added to lines 230-235.

The above two experiments demonstrated that the polymer was indeed engineered at the selected biological sites on the cell surface.

2.2 *The authors assume that by observing a decline in monomer concentration that*

polymerization occurred. This is however not sufficient evidence. Monomer might glue to protein, maybe it has diffused into the cells. Moreover, there is no guarantee that a RAFT process took place. It is known that compounds found in cell growth media can interfere with polymerizations and the authors therefore need to build up their system, supported by molecular weight data via SEC, that this is indeed a controlled process. This might start with polymerization in water using the same concentrations before moving to serum free media as the presence of proteins might not allow SEC analysis.

Response: We have demonstrated the occurrence of polymerization with 2 experiments.

1) Direct characterization of polymers grown on cells.

We anchored CTA containing a cleavable disulfide bond to cells, performed polymerization, cleaved the polymers, and collected the products of multiple parallel polymerization reactions for NMR characterization (Fig. 2B) to demonstrate the formation of polymers. The description has been revised (lines 181-185).

2) Characterization of polymers in the supernatant after cell polymerization.

The polymerization system we usually used was to mix living cells anchored with CTA with a solution containing monomer, free CTA, and a small amount of hydrogen peroxide, and the polymerization was initiated by adding a small amount of ferrous ammonium sulfate. We collected the polymerized solution and used NMR to characterize the polymers produced in the solution by the Fenton-RAFT reaction initiated by free CTA (Fig. S7). The monomer conversion (*Conv.*) was calculated by comparing the integrals of the unreacted vinyl protons (a and a') and the total monomer protons (b) (Fig. S2). We usually used this method to monitor the experimental process because the collection step for cell surface polymer was tedious and required a large number of samples.

We did not verify the occurrence of RAFT by detecting a decrease in monomer concentration.

We investigated whether this polymerization process is controlled in this revision, in water (Fig. S3A) and in culture medium (Fig. S3B), respectively. The results showed the linear increase of the molecular weight with respect to the monomer conversion and low PDI, confirming the chain-growth nature and well-controlled nature of the polymerization reaction. The corresponding descriptions are added to lines 134-135.

2.3 *Is there a way to express the amount of CTA per cells? What happens to this numbers when the cell start proliferating? With each cell division this number should half. Is proliferation affected by labelling, CTA reaction and polymerization? I suppose the processes here need to be carried out quickly before cells change, but it would be useful to learn more about the cell metabolism while the cells are chemically altered.*

Response: We thank to the reviewer for the suggestions! Based on the reviewer's comments, we have supplemented three experiments during this revision.

1) Quantification of cell surface CTA and its changes after cell proliferation.

To measure the average amount of CTA per cell, we incubated Ac₄ManNAz metabolically labeled cells (Cell_{Sia}-azide) with DBCO-PEG₃-F, thereby introducing a fluorine label at the metabolically labeled Sia site. We performed ¹⁹F NMR on lysates from a given number of cells and calculated that on average there are $\sim 2.3 \times 10^9$ F atoms per Cell_{Sia}-azide, *i.e.*, $\sim 2.3 \times 10^9$ azido groups (Fig. S6A).

In the original manuscript, we used DBCO-Cy5 to stain metabolically labeled cells (Cell-azide), and cells with CTA installed (Cell-CTA), respectively (Fig. S6B). The difference in fluorescence signal between these two corresponds to the modification amount of CTA.

Since the fluorescence intensity of a single Cell_{Sia}-azide after DBCO-Cy5 staining corresponds to 2.3×10^9 azido groups, we thus deduced that the average number of CTA per Cell_{Sia}-CTA, Cell_{GalNAc}-CTA, Cell_{Met}-CTA, and Cell_{Cho}-CTA was 4.0×10^8 , 1.4×10^8 , 1.4×10^8 , and 1.4×10^8 , respectively (Fig. S6B).

After 24 h of culture, the average number of CTA per Cell_{Sia}-CTA, Cell_{GalNAc}-CTA, Cell_{Met}-CTA, and Cell_{Cho}-CTA was 3.0×10^8 , 7.0×10^7 , 6.6×10^7 , and 6.7×10^7 , respectively (Fig. S6C). Compared with the data of 0 h of proliferation, it was reduced by 25~50%. The change in the amount of CTA on the cell surface was a result of the dilution of CTA on the cell membrane due to cell proliferation. The corresponding discussion has been added to the revised manuscript (lines 164, 165).

2) Effect of metabolic labeling and CTA installation on cell proliferation

In the original manuscript, we investigated the effect of polymerization at different sites on cell proliferation (Fig. 4B, Fig. 2B of the original manuscript) and found that all four Cell-P proliferated after 72 h. Cell_{GalNAc}-P had a similar proliferative ability as native cells. Cell_{Met}-P, although initially slow to proliferate, had essentially the same cell number as native cells after 72 h of culture. Cell_{Cho}-P proliferated slowly after 24 h. Cell_{Sia}-P proliferated slowly compared to native cells. The corresponding discussion has been added to the revised manuscript (lines 313-316).

In this revision, we have investigated the cell proliferation ability of Cell-azide and Cell-CTA (Fig. S14). Overall, the metabolic labeling had little effect on cell proliferation, while Cell-CTA showed a slight decrease in proliferation compared to native cells, but the overall decrease was less than that of Cell-P. The corresponding discussion has been added to lines 316-320, 346.

3) Effect of cellular modification on metabolism

As suggested by the reviewer, we performed Seahorse XF Cell Mito Stress Test on Cell-azide (Fig. S18A), Cell-CTA (Fig. S18B), and Cell-P (Fig. 4G) to examine the effect on cellular metabolism.

The oxygen consumption rate (OCR) profiles of Cell-azide, Cell-CTA, and Cell-P were similar to those of native cells. The basal respiration capacity of Cell-azide was unaffected by metabolic labeling. The basal respiration levels of Cell_{Sia}-CTA and Cell_{Cho}-CTA were slightly reduced after CTA installation, and Cell_{GalNAc}-CTA and

Cell_{Met}-CTA were not significantly different from native cells. In contrast, the basal respiration levels of Cell-P were reduced after *in situ* polymerization (by 19.7%~35.7%).

After mitochondrial membrane decoupling with the carbonyl cyanide-4 (trifluoromethoxy) phenylhydrazone (FCCP), a robust increase in OCR was observed for Cell-azide, Cell-CTA, and Cell-P. The maximum respiration levels of Cell-azide and Cell-CTA were not statistically different from native cells, while Cell-P was reduced by 22.0~27.0%.

Taken together, these data indicated an excellent basal and maximal respiration capacity of the cells after metabolic labeling and CTA anchoring. While after *in situ* polymerization, the cells exhibited acceptable basal and maximal respiration capacity. The corresponding discussion has been added on lines 366-373.

2.4 *The process here is a grafting from process, which is widely done using RAFT polymerization. However, the researchers usually add free RAFT agent in solution to maintain livingness as during RAFT equilibrium the RAFT agent is lost from the surface and it is unlikely to return if there is no excess RAFT agent. Why was this not done here?*

Response: The reviewer has a very good point. We did add a free RAFT agent to the cell polymerization system. However, this condition is only mentioned in Fig. 1B and the Supplementary Information. To avoid misunderstanding, we have added the relevant content to the caption of Fig. 1.

2.5 *Figure 1C: is it possible that the polymer is simply absorbed on the surface? What does the control look like in the absence of CTA? Figure 5 suggests that the CTA is vital, but I still think it would be useful to think about some controls.*

Response: We thank the reviewer for the suggestions! In the revised manuscript, we have added the scanning electron micrographs of the cells from the samples in which polymerization was initiated in the absence of CTA installation (Fig. 2A). The morphology of the cells is similar to native cells and significantly different from the cells grown with polymer. We did not observe any non-specific adsorption of the polymer to the cell surface. The corresponding content has been added to lines 177-180.

2.6 *I am trying to understand how the polymer was cleaved from the surface. The polymerization was carried out in RMPI media. How exactly were the polymers cleaved and purified in order to obtain such nice NMR spectra. I would have thought that there are plenty of impurities, if not from the cell growth media then from the molecules excreted from the cells.*

Response: To cleave the polymer, we prepared a chain transfer agent (DBCO-SS-BTPA) containing a disulfide bond and performed the polymerization step to produce Cell_{Sia}-

P. We carefully washed Cells_{sia}-P with an incomplete culture medium and then separated the polymerization product from the cells by cleaving the disulfide bonds with tris(2-carboxyethyl)phosphine hydrochloride (TCEP). We centrifuged the mixture at 300 g for 5 min, collected the supernatant, and then dialyzed it in water for 48 h using dialysis tubes (MWCO=1 K). We collected the products of several parallel reactions, dissolved them in D₂O, and performed ¹H NMR characterization (Fig. 2B). Since we did not lyse the cells and the TCEP cleavage time was short, the number of free biomolecules from the cells was small compared to the polymer products and had a limited impact on the ¹H NMR results (small impurity peaks were observed at 2.12 and 2.53 ppm). We have added the corresponding description (lines 181-185).

2.7 The authors used access of WGA as a means to confirm the chain extension. Is this really a valid approach? All it means is that more polymer is on the surface, but this can also be the case when more polymer is adsorbed. Glycans on the cell surface can form strong H-bonding that can retain some polymer. I think the authors need to add a whole suite of experiments to prove this such as increase in fluorescence intensity as in Figure 5 using appropriate controls. Alternatively this part can be fully removed.

Response: We thank the reviewer for the suggestion and we have removed these data (lines 190-193).

2.8 Line 212: why would a polymer have a brush regime? This is usually observed when the grafting density is very high.

Response: Thank you very much for pointing this out. We originally intended to describe the polymer as having a "chain structure". We have changed this description in the revised manuscript (lines 381-382).

2.9 Figure 2: How does CD3 access change after days of proliferation? The sites should be easier to access as there are less polymers on the surface as cells grow in number while the polymer amount is still the same. Figure 3 f-h: It is almost possible to group the polymer b and c together as well as d and e. The authors argues that b and d protect the lectin binding sites. Could it also just be that different amounts of polymers are attached to the surface in addition to different anchorage sites? Figure 5 suggest that there is more polymer on the surface with GalNAc.

Response: As suggested by the reviewer, we cultured Cell-P for 48 h and then examined the binding to anti-CD3-FITC. There was no significant difference in the antibody binding signal between Cell-P and native cells regardless of the polymer growth site (Fig. S20). The reason for this is presumably that cell proliferation reduces the density of polymers on the cells so that they can not interfere with antibody binding. We have added the corresponding description in the revised manuscript (lines 386-391).

We have grouped the polymers b and c together as well as d and e in Fig. 5.

In the original manuscript, we considered the effect of the number of polymer growth sites on the results, so we adjusted the CTA number on GalNAc, Met, and Cho sites to be the same. In the revised manuscript, we have quantified the number of CTA modified at different sites (Fig. S6), which further confirms our original design and experimental results. Since the number of CTA at GalNAc, Met, and Cho sites is consistent, we believe that the difference in the blocking effect against lectins is due to differences in the polymer growth sites.

2.10 *Figure 5: This is an important experiment and should have been discussed much earlier. It clearly shows the presence of polymer on the surface and it appears that there is no polymer in the absence of CTA. However, there is no experimental part that describes the experiment in detail such as the amount of functional monomer added. Also were the polymers washed before imaging? Why is there no fluorescent polymer in solution as the Fenton reagents would have started a free radical polymerization.*

Response: As suggested by the reviewer, we have moved Fig. 5 forward to Fig. 2C and Fig. 3A~F.

The cells were washed three times after the SSP polymerization operation and after incubation with fluorescent reagents so that the fluorescence of the free polymer does not appear in the images.

2.11 *Figure 5A: labelling with AA-PEF-azide: I am not too sure what these experiments show as the discussion in the text is not clear. The authors use cells with and without CTA, but then did not do any polymerization. There are no fluorescent signals on both cells. What did the authors show here?*

Response: For Fig. 5A in the original manuscript (Fig. 2C in the revised manuscript), we have revised the relevant discussion (lines 210-230), which has been specified in the response to 2.1. In brief, the two non-polymerized samples, with and without CTA installation, are used to preclude the adsorption possibility of fluorescent reagent (SA-Cy3) on the cell surface.

2.12 *Figure 5F: I noticed some uneven distribution of polymer on Cho. Why is that?*

Response: We speculate that it is possible that after a period of culture, Cell_{Cho}-P appears to phase separate under the influence of polymer modifications.

Reviewer #3

3.0 In this manuscript the authors report on their development and investigation of a method to prepare polymeric structures on the membrane of living cells. The manuscript is well written and the data is well presented. I really enjoyed reading it and think that the presented results are certainly of high significance to the field.

However (...there is always a "but"... feeling sorry for being the one raising concerns as I consider the authors' findings to be quite exciting), there are some remaining issues/questions I need to point out, the major ones being the following two:

Response: We are very grateful to the reviewer for the high opinion of our work.

3.1 *According to the methods section, bioorthogonal azide cyclooctyne ligation on live cells was done at a concentration of 40 μM DBCO-BTPA for 1 hour. Considering the second-order rate of this reaction (known to be $<1 \text{ M}^{-1} \text{ s}^{-1}$) this would result in a click-conversion of only $\sim 10\%$. Hence, most of the metabolically installed azide-tags are not used for further modification. Under these conditions it would actually take >1 day for the reaction to go to completion (99%), and ~ 5 hours to reach 50%. Even more striking, the reaction as described for "Visualization of copolymers on living cell surface" (10 μM DBCO, 30%) would result in a conversion of only $\sim 1\%$. In both cases it might be even less considering a reaction temperature of 4 $^{\circ}\text{C}$. I'd thus recommend the authors to comment on reaction kinetics and why these specific conditions have been chosen. After all, this affects the construction of polymers on the membrane, but also quantification by fluorescence measurements.*

Response: We thank the reviewer for the thoughtful comments. Regarding the reaction kinetics, we propose that the labeling of cell surface azides with DBCO-BTPA was a heterogeneous reaction, rather than a two-component homogeneous solution reaction. The local concentration of azides on the cell membrane was considerably high, whereas DBCO-BTPA in solution was in large excess, allowing a significantly accelerated labeling process. According to the previous literature (*Angew. Chem. Int. Ed.* **2008**, *47*, 2253), the treatment with 30 μM dibenzocyclooctyne (DIBO) at 4 $^{\circ}\text{C}$ for 1 h was able to modify approximately 70% of the cell membrane. Other references have also reported similar labeling conditions using different cyclooctynes, further indicating the suitability of these conditions for cell surface modification (*Proc. Natl. Acad. Sci. U. S. A.* **2007**, *104*, 16793, *Angew. Chem. Int. Ed.* **2010**, *49*, 9422, *J. Am. Chem. Soc.* **2010**, *132*, 3688, *Angew. Chem. Int. Ed.* **2012**, *51*, 920, *Proc. Natl. Acad. Sci. USA* **2014**, *111*, 5456, *Nat. Microbiol.* **2017**, *2*, 17099, et al.). Considering that DBCO has been reported to be significantly more reactive than DIBO ($0.31 \text{ M}^{-1} \text{ s}^{-1}$ vs. $0.07 \text{ M}^{-1} \text{ s}^{-1}$, *Chem. Commun.* **2010**, *46*, 97 and *ChemBioChem* **2011**, *12*, 1912, respectively), we chose to use the current labeling conditions for cell labeling, confident that they should result in

the modification of a sufficient amount of azido groups on the cell membrane.

In the revised manuscript, we added the reference (Ref. 65) for the experimental conditions of the click reaction to the experimental section.

3.2 *The authors have carefully chosen - based on preliminary experiments - to perform polymerization for 2 min. However, I'd recommend to avoid saying that the polymerization is "complete" or "completed in 2 min".*

More importantly, under these conditions the authors analyzed the on-cell formed polymer and describe a PDI of 1.37 (together with further data in Fig. S6). It seems to me that the authors have conducted all further polymerizations in the same way, which would mean that there is no data on different polymer lengths/sizes. Considering further investigations, in particular when size (or point of attachment) seems to matter (cf. Fig. 3I), I think it's crucial to understand and determine the morphology/size of the polymer corona (at least via modeling). Without such data it's difficult to make claims such as the one related to the data shown in Fig. 3I. Moreover, the images provided in Fig. 5B indicate, as the authors describe it in the main text, the formation of "clusters on the surface of living cells". While this might be related to the described copolymerization of functional monomers, it further raises questions about the morphology of the polymer corona.

In addition, it would be interesting to see the results of a few selected experiments when trying different reaction conditions / polymerization times (thus varying polymer sizes). For instance, what is the effect on lectin binding if Cell(Sia)-P and/or Cell(GalNAc)-P are prepared with reduced polymerization time?

Could reduced lectin binding also be a result of structural changes of the glycans rather than steric hindrance? In that sense, please add data for lectin binding of metabolically labeled but not polymerized cells as a control.

Response: Thank you for your suggestion! we have removed all descriptions of "complete" or "completed in 2 min".

In the revised manuscript, we shortened the polymerization time to 1 min, prepared a lower molecular weight polymer, investigated the effect of the lower molecular weight on polymer morphology, lectin binding, and cell membrane retention time, and characterized the morphology of the polymer on the cell using stimulated emission depletion (STED) microscope. The detailed results are presented below:

1) Effect of reduced polymerization time

We copolymerized HPMA and AM-PEG₄-N₃ monomers at the GalNAc, Met and Cho sites with a polymerization time of 1 min, respectively. Using DBCO-Cy5 staining, we characterized the cells after polymerization by confocal laser scanning microscope (CLSM) and flow cytometry (FCM), respectively (Fig. S11A-D). The fluorescence signal was reduced compared to the corresponding sample polymerized for 2 min, indicating that the polymer was successfully grown but had a shorter chain length.

Cell-P^{azide} produced by 1 min polymerization was stained with DBCO-Cy5 and then cultured for 24 h. The percentage of retention on the cell membrane was slightly lower than that of the cell sample polymerized for 2 min, but the ordering relationship between sites was unchanged (Fig. 3F). The corresponding description has been added to lines 268-270.

We also prepared Cell-P with different polymer lengths by polymerizing HPMA at the Sia and GalNAc sites for 1 and 2 min, respectively. Using the lectin S-WGA as a model, all Cell-P prepared using different polymerization times blocked lectin binding, and the degree of blocking was positively correlated with the polymer length (Fig. S26).

Since the lectin binding levels for Cell-azide and Cell-CTA are similar to those for native cells (Fig. S26), we hypothesize that the reduction in lectin binding was due to structural changes in the polymer rather than the sugar. The corresponding description has been added on lines 471-477.

2) STED-based observation of macromolecular morphology

To better observe the morphology of the polymer on the cell membrane, we performed STED imaging of Cell_{GalNAc}-P^{azide}, Cell_{Met}-P^{azide}, and Cell_{Cho}-P^{azide} using Click-iT™ sDIBO as the fluorescent signal molecule (Fig. 3G). The polymer was visible in the STED images distributed in clusters on the cell membrane, and when the polymerization time was extended, the fluorescent clusters were larger and more clearly delineated compared to the background. We also incubated the stained cells for another 24 h and found that the cell surface polymers still appeared in clusters, but with weaker intensity than in the samples imaged immediately after polymerization, and the fluorescence signal appeared inside the cells (Fig. 3H). After 24 h of incubation, cells with longer polymerization times also showed larger fluorescent clusters. For comparison, we also observed Cell-azide with STED and found that, regardless of the sites of azide tag introduction, the cell membranes showed a continuous distribution of weak fluorescence (Fig. S13), which was very different from that of polymer-grafted cells. The corresponding description has been added on lines 274-277, 294-307.

Minor comments:

3.3 Please indicate that CTA installation is done via "strain-promoted azide alkyne cycloaddition (SPAAC)" (in the text and Fig. 1A). Please use "azido-functionalized molecules", "azide tags", "azides", etc. and "click-tags" or "clickable tags" (or similar) rather than "N3" and "anchor sites".

Response: We have revised the description in the revised manuscript.

3.4 I agree that a viability of 87% (Fig. S1) is promising, but it doesn't match the authors' statement "essentially no effect on cellular activity" (actually "viability").

Response: We are grateful to the reviewer for pointing this out. We have revised the

relevant description (lines 127-128).

3.5 *Please quantify the data as shown in Fig. S3B. It seems there are at least a few percent of dead cells. Certainly no issue, but providing a number seems more accurate than saying "largely maintained".*

Response: The percentages of living cells have been added to line 136.

3.6 *The authors show that modified cells are still able to proliferate. What happens to the polymer corona when cells divide?*

How do daughter cells look like / behave?

While this would probably require even more additional experiments (very likely too many to reasonably ask for), the authors should at least comment on this and/or discuss.

Response: We performed CLSM imaging in the original manuscript for Cell-P^{azide} 24 h after polymerization. The retention ratios (RR) of polymers at different sites of polymerization were: RR_{GalNAc-P^{azide}} (76.3%), RR_{Met-P^{azide}} (61.0%), and RR_{Cho-P^{azide}} (53.0%) (Fig. 3F). The reduction of polymers on the cell membrane can be attributed to two reasons: 1) dilution of the polymers due to cell proliferation and 2) endocytosis of the polymer-modified biomolecules into the cells (fluorescent signal could be seen inside the cells) or exocytosis to the outside of the cells.

In the revised manuscript, we further observed the morphology of Cell-P^{azide} after 24 h culturing using STED (Fig. 3H). It was found that the polymers on the cell surface still appeared in clusters, but with weaker intensity than in the samples imaged immediately after polymerization, and the fluorescence signal appeared inside the cells.

The corresponding description has been added on lines 258-263, 297-300.

3.7 *Fig. 3: While it's more or less obvious when looking at the whole figure (in particular 3E), please indicate the cell types a-e in the figure caption. Same for Fig. 2 (e.g., when looking at 2F only without immediately noticing the legend in 2B).*

Response: We are grateful to the reviewer for the suggestion. We have revised these figures (Figs. 4 and 5).

3.8 *Do the authors have any idea about the "density" of metabolically incorporated azide tags on the cell membrane? ...also referring to the "modification amount of CTA" as described by the authors.*

Response: To measure the average amount of CTA per cell, we incubated Ac₄ManNAz metabolically labeled cells (Cells_{Sia}-azide) with DBCO-PEG₃-F, thereby introducing a fluorine label at the metabolically labeled Sia site. We performed ¹⁹F NMR on lysates

from a given number of cells and calculated that on average there are $\sim 2.3 \times 10^9$ F atoms per Cell_{Sia}-azide, *i.e.*, $\sim 2.3 \times 10^9$ azido groups (Fig. S6A).

In the original manuscript, we used DBCO-Cy5 to stain metabolically labeled cells (Cell-azide), and cells with CTA installed (Cell-CTA), respectively (Fig. S6B). The difference in fluorescence signal between these two corresponds to the modification amount of CTA.

Since the fluorescence intensity of a single Cell_{Sia}-azide after DBCO-Cy5 staining corresponds to 2.3×10^9 azido groups, we thus deduced that the average number of CTA per Cell_{Sia}-CTA, Cell_{GalNAc}-CTA, Cell_{Met}-CTA, and Cell_{Cho}-CTA was 4.0×10^8 , 1.4×10^8 , 1.4×10^8 , and 1.4×10^8 , respectively (Fig. S6B).

After 24 h of culture, the average number of CTA per Cell_{Sia}-CTA, Cell_{GalNAc}-CTA, Cell_{Met}-CTA, and Cell_{Cho}-CTA was 3.0×10^8 , 7.0×10^7 , 6.6×10^7 , and 6.7×10^7 , respectively (Fig. S6C).

The corresponding discussion has been added to the revised manuscript lines 164-165).

3.9 *Sugar monomers have been prepared starting from azido-functionalized protected monosaccharides via copper-catalyzed click chemistry. Is there any specific reason for this strategy, for instance, in contrast to Staudinger reduction of the azide to the respective amines followed by direct attachment of the methacryloyl chloride to avoid the triazole being part of the structure? It might act as a linker/spacer between the sugar moieties and the polymer backbone, but that's hard to say without any comparison.*

Response: [REDACTED]

REVIEWER COMMENTS

Reviewer #1 (Remarks to the Author):

The authors addressed the most questions , but that is not enough, the reviewer thinks.

Regarding 1.2, the retention time of the polymer is not well responded. The reviewer asked how long the polymer existed on the cell surface.

Regarding 1.4, the Fenton reaction or reactive oxygen speices are general problem to cells, but I wonder why the cells had no impact. Are there any quenchers in culture medium, such as ascorbic acid, cysteins, etc. Authors are asked to clarify this issue.

Regarding 1.8, the evidence of selective polymerization at N3 on cell surface is still vague. if we look the Fig 3C where NEU treatment was done, there is still fluorecence observed even after the treatment. how would the authors answer the issue?

I was asked to check the response to Reviewer 2 by [the editor], so I wrote down my comments here.

Regarding 2.1, the reviewer 2 thought it is difficult to find the new finding from this manuscript. The authors could not respond to this point correctly.

Regarding 2.5, SEM is not optimal to address this issue. We need labelled polymers and confocal microscopy and flow cytometry.

Regarding 2.12, the authors did not address this issue. We cannot exclude the possibility of eneven modification. The reviewer do not think the phase separation since there is no driving force for that.

Reviewer #3 (Remarks to the Author):

The authors have carefully revised their manuscript and added a further data to support their findings and claims. All my previous comments/concerns have been addressed, but there is one remaining issue that I feel needs clarification (see below). Nevertheless, in agreement with my previous evaluation I consider this study to be of high significance and fully support publication in Nature Communications.

In their response to comment 3.1 the authors "propose that the labeling of cell surface azides with DBCO-BTPA was a heterogeneous reaction, rather than a two-component homogeneous solution reaction. The local concentration of azides on the cell membrane was considerably high, whereas DBCO-BTPA in solution was in large excess, allowing a significantly accelerated labeling process." However, the high local concentration of azides has no effect on the rate of conversion (percentage) of

these tags. In other words: the reaction of one azide doesn't depend on how many azides are in close proximity. In fact, such a heterogeneous reaction can be described with pseudo-first order kinetics. Here, the azide concentration doesn't play any role, while the lower DBCO concentration can (paradoxically) be considered constant (due to the mentioned excess in solution). Hence, the second order rate constant can be combined with the DBCO concentration to a "new" constant and the reaction kinetics can be treated as a first order process (again: in which the azide concentration has no effect on the overall %conversion). Using a rate constant of $0.31 \text{ M}^{-1}\text{s}^{-1}$ (as described by the authors in their response) the azide conversion after a reaction time of 1 hour is approximately only ~3%. Hence, being a heterogeneous reaction cannot be taken as the sole argument to explain a higher conversion. In addition, I was not able to find any comment in the cited reference (Angew. Chem. Int. Ed. 2008, 47, 2253) regarding a conversion of 70% within 1 hour (but might have missed it). In Fig. 3 of this paper the fluorescent signal is still increasing after a reaction time of 3 hours.

The authors mention several other references/studies in which similar concentrations have been used and I agree that those reaction conditions can be considered "suitable", in particular for imaging (as long as there is sufficient signal intensity). However, when aiming for modification of most of the installed azide tags, I'd expect longer reaction times to be beneficial.

I'd thus appreciate if the authors could further comment on the selected concentrations and reaction times.

Manuscript number: NCOMMS-22-52827A

Response to the reviewers' comments

Reviewer #1

1) Regarding 1.2, the retention time of the polymer is not well responded. The reviewer asked how long the polymer existed on the cell surface.

Response: We are extremely grateful to the Respected Reviewer for the insightful comment and suggestion. Indeed, the retention time of polymers on the cell membrane is a key parameter for the cell surface engineering. In line with this insightful suggestion, we have carefully investigated the retention time of polymers on the cell surface, and the manuscript has been revised with the results supplemented (Fig. S15, Lines 311~318).

We have copolymerized *N*-(2-hydroxypropyl)methacrylamide (HPMA) and acrylamide-poly (ethylene glycol)₄-azide (AA-PEG₄-azide) at different sites (GalNAc, Met, and Cho) on the cells to generate Cell-P^{azide}, and monitored the retention profiles of respective polymers.

Thanks to the Respected Reviewer's insightful suggestion, an important finding has been acquired: the retention time of the polymer at the glycan site (GalNAc) on the cell membrane is much longer than that of the polymers at the protein (Met) and lipid (Cho) sites. It is also important to note that cell membrane stability of the polymers at Cho sites is likely to be superior to polymers inserted in the cell membrane through hydrophobic interactions. Taken together, this site-specific variation of polymer properties highlights the importance of site-selected *in situ* polymerization for living cell surface engineering and promises it as a generally applicable, powerful tool for studying cell surface polymer dynamics and associated biological consequences.

2) Regarding 1.4, the Fenton reaction or reactive oxygen species are general problem to cells, but I wonder why the cells had no impact. Are there any quenchers in culture medium, such as ascorbic acid, cysteine, etc. Authors are asked to clarify this issue.

Response: We are extremely grateful to the Respected Reviewer for the insightful comment and suggestion, which have enabled us to provide comprehensive additional experimental confirmation, elaborate more clearly on reactive oxygen species (ROS), and thoroughly refine the content of the manuscript.

The polymerization conditions have been extensively surveyed and optimized to minimize the effect of ROS, as confirmed by cell viability and proliferation assays (Figs. 4A, B, S4A~C), F-actin staining (Fig. 4C, D), ROS imaging (Fig. S18), DNA and cytoplasmic protein assays (Fig. 4E, F), cell function test (Fig. 4G~J).

The effect of ROS on cells is mainly dependent on the location, environment, and exposure time. The minimized effect of ROS observed herein under

optimized polymerization condition is due to the following:

1) As mentioned by the Respected Reviewer, the polymerization medium contains 1 $\mu\text{g}/\text{mL}$ reduced glutathione; also, after polymerization, the RPMI-1640 complete culture medium used to terminate the reaction contains 1 mM sodium pyruvate (a free radical scavenger). These two components can ensure the minimization of ROS effect on cells under the optimized polymerization condition used herein.

2) The polymerization reaction proceeds at the cell surface and within a short time duration of 2 min. There is highly likely only a very limited number of ROS, if any, that can potentially enter the cells.

3) Cells by themselves naturally have several antioxidant defense systems, including protein-based enzymatic scavengers (e.g., peroxidase); cells can also deal with ROS through a variety of non-enzymatic pathways (e.g., with the help of glutathione).

Thanks to the Respected Reviewer's insightful suggestion, we have been able to clarify the minimized effect of ROS on cells in the revised manuscript (Lines 412~416).

3) Regarding 1.8, the evidence of selective polymerization at N_3 on cell surface is still vague. if we look the Fig 3C where NEU treatment was done, there is still fluorescence observed even after the treatment. how would the authors answer the issue?

Response: We are extremely grateful to the Respected Reviewer for the insightful comment and suggestion. The insightful comment has prompted us to elaborate more comprehensively and clearly, both experimentally and in the manuscript content, on site-selected polymerization. Indeed, in retrospect, we have not been able to illustrate clearly in the last round of revision. The sialic acid (Sia) at the end of the cell surface glycan chains can be attached to the glycan at the penultimate position via an $\alpha 2-3$, $\alpha 2-6$ or $\alpha 2-8$ glycosidic linkage. The commercial sialidase (NEU) used herein (from *Clostridium perfringens*) is catalytically active toward all three types of linkages; however, sialidases in general exhibit different cleavage activities for different glycosidic linkages (specifically, the relative rates for NEU herein are in the order of $\alpha 2-3 > \alpha 2-6 > \alpha 2-8$). Therefore, cell surface Sia can only be partially cleaved at maximum by NEU (Fig. 2D). This incomplete cleavage has been extensively observed and reported in the literature. Our group has accumulated years of experience in using NEU to cleave Sia, and in an independent experiment, NEU is identified to be capable of cleaving 68.6% of Sia from a natural cell surface (Fig. S11); this cleavage efficiency is comparable to that observed for $\text{Cell}_{\text{Sia}}\text{-P}^{\text{biotin}}$ (66.5%) (Fig. 2D). This demonstrates the selective polymerization from the Sia site. We have revised the manuscript by incorporating a more clear illustration of the site-selected polymerization (Lines 230~232).

4) Regarding 2.1, the reviewer 2 thought it is difficult to find the new finding from

this manuscript. The authors could not respond to this point correctly.

Response: We are extremely grateful to the Respected Reviewer for the insightful comment and suggestion. The insightful comment has prompted us to elaborate more comprehensively and clearly, both experimentally and in the manuscript content, on site-selected polymerization from the cell surface. Indeed, in retrospect, we have not been able to illustrate clearly in the last round of revision.

Comprehensive experiments and lines of evidence support the site-selected polymerization from the cell surface. The growth of polymer from the cell surface is demonstrated via the following: only with the initial attachment of chain transfer agent (CTA) to the Sia site and subsequent polymerization (with HPMA and AA-PEG₄-biotin), has polymer been observed on cell surface, as confirmed by confocal imaging (Fig. 2C), flow cytometry (Fig. S10), and scanning electron microscopy (Fig. 2A); without CTA attachment, no polymer has been observed under otherwise identical condition; further, with the omission of either polymerization or both CTA attachment and polymerization, no polymer has been identified. For this experimental proof, the Respected Reviewer 2 has also affirmed in the first round of revision (Comment 2.10). The growth of polymer at the specific site is demonstrated via the following: the selective cleavage of Sia by NEU releases the polymer from the cell surface. This has been comprehensively elaborated in the response to Comment 3.

As a demonstration of the significance of our site-selected *in situ* polymerization for living cell surface engineering, we experimentally found that the growth site of the polymer affects the nature of the polymer itself and the functions it performs, as manifested in the following: 1) the retention time of the polymers at different sites is different, with a longer value from the glycan site; 2) the polymer grown at the glycan site exhibits a stronger blocking effect on the lectin recognition of cells; 3) most significantly, for the first time a glycopolymer has been grown *in situ* at the end of the cell's natural glycan chain, generating a biomimetic glycocalyx that allows the engineered cells to be recognizable by lectins (like the natural glycocalyx), but resistant to aggregation (unlike the natural glycocalyx).

We are extremely grateful to the Respected Reviewer for the insightful comment and suggestion, which has enabled us to elaborate more clearly on the innovative aspect of the manuscript and greatly improved the scientific quality of the manuscript.

Taken together, the above summarized new finding highlights the importance of site-selected *in situ* polymerization for living cell surface engineering and promises it as a generally applicable, powerful tool for studying cell surface polymer dynamics and associated biological consequences.

5) Regarding 2.5, SEM is not optimal to address this issue. We need labelled polymers and confocal microscopy and flow cytometry.

Response: We are extremely grateful to the Respected Reviewer for the insightful comment and suggestion. We have additionally provided data based on confocal microscopy (Fig. 2C) and flow cytometry (Fig. S10). Only with the initial attachment of CTA to the Sia site and subsequent polymerization, has polymer been observed on cell surface. Without CTA attachment, no polymer has been observed under otherwise identical condition. Further, with the omission of either polymerization or both CTA attachment and polymerization, no polymer has been identified. These results are consistent with those obtained by the electron microscopy (Fig. 2A) and together demonstrate the cell surface polymer growth.

6) Regarding 2.12, the authors did not address this issue. We cannot exclude the possibility of uneven modification. The reviewer does not think the phase separation since there is no driving force for that.

Response: We are extremely grateful to the Respected Reviewer for the insightful comment and suggestion, which have prompted an in-depth understanding of the cell surface polymerization process.

The cell surface after metabolic labeling is observed to exhibit a uniform distribution of small-molecule species (Fig. S14A). This is consistent with the extensively reported literature observations. This uniform distribution is also supposedly applicable to small-molecule CTA attachment. The uneven distribution of polymers on the cell surface most likely occurs at the polymerization stage, instead of via the post-polymerization phase separation. Indeed, the fixed cells undergoing an otherwise identical polymerization process exhibit an uneven distribution of polymers (Fig. S9). Proximity-triggered cooperative polymerization has been observed previously to be a mechanism to preferentially generate additional polymers adjacent to an existing polymer chain. This can result in the clustering of polymer chains and uneven distribution of polymers on the cell surface. We have revised the mechanistic proposal in the revised manuscript accordingly (Lines 305~310). In summary, we are really grateful to the Respected Reviewer for patiently and carefully perusing through the manuscript over and over again and providing extremely insightful comments and suggestions. These comments and suggestions have enabled in-depth elaboration, both experimentally and in the manuscript content, on the importance of the site-selected *in situ* polymerization for living cell surface engineering approach described herein. With the significant improvement on quality, we believe that the revised manuscript should appeal to the diverse readership targeted by *Nat. Commun.* and therefore be worthy of publication in this journal.

Reviewer #3:

The authors have carefully revised their manuscript and added a further data to support their findings and claims. All my previous comments/concerns have been addressed, but there is one remaining issue that I feel needs clarification (see below). Nevertheless, in agreement with my previous evaluation I consider this study to be of high significance and fully support publication in Nature Communications.

In their response to comment 3.1 the authors "propose that the labeling of cell surface azides with DBCO-BTPA was a heterogeneous reaction, rather than a two-component homogeneous solution reaction. The local concentration of azides on the cell membrane was considerably high, whereas DBCO-BTPA in solution was in large excess, allowing a significantly accelerated labeling process."

However, the high local concentration of azides has no effect on the rate of conversion (percentage) of these tags. In other words: the reaction of one azide doesn't depend on how many azides are in close proximity. In fact, such a heterogeneous reaction can be described with pseudo-first order kinetics. Here, the azide concentration doesn't play any role, while the lower DBCO concentration can (paradoxically) be considered constant (due to the mentioned excess in solution). Hence, the second order rate constant can be combined with the DBCO concentration to a "new" constant and the reaction kinetics can be treated as a first order process (again: in which the azide concentration has no effect on the overall %conversion). Using a rate constant of $0.31 \text{ M}^{-1}\text{s}^{-1}$ (as described by the authors in their response) the azide conversion after a reaction time of 1 hour is approximately only ~3%. Hence, being a heterogenous reaction cannot be taken as the sole argument to explain a higher conversion. In addition, I was not able to find any comment in the cited reference (Angew. Chem. Int. Ed. 2008, 47, 2253) regarding a conversion of 70% within 1 hour (but might have missed it). In Fig. 3 of this paper the fluorescent signal is still increasing after a reaction time of 3 hours.

The authors mention several other references/studies in which similar concentrations have been used and I agree that those reaction conditions can be considered "suitable", in particular for imaging (as long as there is sufficient signal intensity). However, when aiming for modification of most of the installed azide tags, I'd expect longer reaction times to be beneficial.

I'd thus appreciate if the authors could further comment on the selected concentrations and reaction times.

Response: We are grateful for the detailed comment and have benefited from the reading. It serves as an important guide for our group in the future to design and optimize click reaction-related experiments. We very much agree with the suggestion that without affecting the cellular activity, it is possible to increase

the concentration of DBCO molecules, raise the temperature, and prolong the reaction time to obtain a higher modification ratio of azide tags. This is not only beneficial for revealing the mechanisms by which engineering operations affect cells at the molecular and cellular levels, but can also facilitate the regulation of engineered cells in a wider range. In line with the suggestion, we have commented on the selected concentrations and reaction times (Lines 658~660).

REVIEWER COMMENTS

Reviewer #1 (Remarks to the Author):

No further comments

Reviewer #3 (Remarks to the Author):

The authors commented on my last concern. However, the addition to the revised manuscript in the methods section is not fully convincing. Without any data or reference linked to such claims (lines 658-660) it can be considered unsubstantiated.

The conversion of installed azide tags at the cell surface is a crucial parameter for the entire process and it would be desirable to see data of selected further experiments (longer reaction time, higher concentration) or, at least, a more in-depth discussion of the click conversion and its implications in the manuscript. The authors could test for remaining azide tags and their accessibility, for instance, by reaction with a fluorescent cyclooctyne and monitoring via fluorescence microscopy (comparing cells with and without the polymer). While quantification (signal intensity) at the end of the experiment can be difficult, monitoring the kinetics (rate of conversion) would directly indicate different azide concentration/availability independent from the signal intensity.

Manuscript number: NCOMMS-22-52827B

Response to the reviewer' comments

Reviewer #3

The authors commented on my last concern. However, the addition to the revised manuscript in the methods section is not fully convincing. Without any data or reference linked to such claims (lines 658-660) it can be considered unsubstantiated.

The conversion of installed azide tags at the cell surface is a crucial parameter for the entire process and it would be desirable to see data of selected further experiments (longer reaction time, higher concentration) or, at least, a more in-depth discussion of the click conversion and its implications in the manuscript. The authors could test for remaining azide tags and their accessibility, for instance, by reaction with a fluorescent cyclooctyne and monitoring via fluorescence microscopy (comparing cells with and without the polymer). While quantification (signal intensity) at the end of the experiment can be difficult, monitoring the kinetics (rate of conversion) would directly indicate different azide concentration/availability independent from the signal intensity.

Response:

We are extremely grateful to the Respected Reviewer for the insightful comments and suggestions. We fully agree with the Respected Reviewer that the degree of conversion for the click reaction is crucial for the entire process and as such we really appreciate the emphasis of this critical point by the Respected Reviewer. In accordance with the Respected Reviewer's suggestions, we have supplemented comprehensive experimental investigation on the azide tags and conversion chemistry. The effects of CTA click reaction conditions on CTA installation amount, cell viability, and polymer synthesis have been elucidated. This has further enabled the understanding of the proposed site-selected polymerization protocol, and provides a very important guiding framework for the continued research of cell surface polymer engineering in our laboratory.

We fully agree with the Reviewer's inspiring comment "The conversion of installed azide tags at the cell surface is a crucial parameter for the entire process". Increase of reagent concentration, extension of reaction time and increase of temperature are commonly used effective ways to improve reaction efficiency. In the field of cell surface engineering, the maintenance of cell integrity needs to be considered when optimizing the engineering strategy. Therefore, we first investigated the effects of increasing DBCO-BTPA (clickable CTA) concentration and click reaction time on cell integrity. A reaction temperature at 4 °C is preferred due to the hydrophobic nature of DBCO-BTPA structure. The cell integrity can be maintained for 1-h click reaction with 40 μM and 100 μM DBCO-BTPA concentrations (azido-sialic acid labeled cells, highest extent of azide installation, Fig. S6). The compatibility of a higher 100 μM concentration indeed fully supports the Respected Reviewer's suggestion on the ability to increase the conversion of azide and degree of polymerization (*vide infra*).

We used SERS (Surface-Enhanced Raman Spectroscopy) to directly test the conversion of azide on the cell surface after DBCO-BTPA treatment (Fig. S7). The vibrations of azide are located in the Raman-silent region of the cells (about 1800 to 2800 cm^{-1}), allowing convenient quantification of azide. We treated Cell_{Sia}-azide, Cell_{Sia}-CTA ($c_{\text{DBCO-BTPA}} = 40 \mu\text{M}$), and Cell_{Sia}-CTA ($c_{\text{DBCO-BTPA}} = 100 \mu\text{M}$) of the same number with sialidase (NEU, the cleavage ratio on different cell surface can be regarded as constant) to release the sialic acids (both azido-labeled and unlabeled) at the end of glycan chains on the cell surface. After the supernatants were collected by centrifugation, the azide signal intensity ($\sim 2120 \text{cm}^{-1}$) was detected by SERS. The order of the accessible azide on the cell surface was Cell_{Sia}-azide > Cell_{Sia}-CTA ($c_{\text{DBCO-BTPA}} = 40 \mu\text{M}$) > Cell_{Sia}-CTA ($c_{\text{DBCO-BTPA}} = 100 \mu\text{M}$). The azide labeling ratio for the groups obtained with $c_{\text{DBCO-BTPA}}$ of 40 and 100 μM were 26.7% and 37.3%, respectively. Therefore, it is proved that increasing DBCO-BTPA concentration is beneficial to obtain a higher azide conversion ratio.

Finally, we studied the effect of increasing azide conversion on polymerization (Fig. 2E). After reacting Cell_{Sia}-azide with DBCO-BTPA of 40 μM or 100 μM for 1 h, *in situ* copolymerization of *N*-(2-hydroxypropyl) methacrylamide (HPMA) and acrylamide-poly (ethylene glycol)₄-biotin (AA-PEG₄-biotin) was performed. The yielded biotin-containing polymer grown on cell surface was indicated by streptavidin-cyanine3 (SA-Cy3) staining. The fluorescence signal of Cell_{Sia}-P^{biotin} ($c_{\text{DBCO-BTPA}} = 100 \mu\text{M}$) was significantly higher than that of Cell_{Sia}-P^{biotin} ($c_{\text{DBCO-BTPA}} = 40 \mu\text{M}$). It was proved that the degree of polymerization engineering was enhanced by increasing the amount of chain transfer agent on the cell surface through increasing the concentration of DBCO-BTPA.

The supplemented experimental results have been added (Figs. S6, S7 and 2E) and the relevant discussion has been added/modified in the text (Lines 160-175; 247-250; 674-676).

Taken together, we wish to express our tremendous gratitude to the Respected Reviewer for providing really insightful comments and suggestions. These comments and suggestions have enabled the substantial improvement of the quality of the manuscript. The careful and patient reading and re-reading of the manuscript by the Respected Reviewer is really admirable and we really appreciate the precious time and dedicated effort. With the significant improvement of manuscript quality, we believe that the revised manuscript should appeal to the diverse readership targeted by *Nat. Commun.* and therefore be worthy of publication in this journal.

REVIEWERS' COMMENTS

Reviewer #3 (Remarks to the Author):

The authors have performed additional experiments and revised the manuscript accordingly, overall fully addressing my previous and last concern.

I thus recommend publication in Nat. Commun. and like to congratulate the authors on this work!

Manuscript number: NCOMMS-22-52827C

Response to the reviewer' comments

Reviewer #3

The authors have performed additional experiments and revised the manuscript accordingly, overall fully addressing my previous and last concern. I thus recommend publication in Nat. Commun. and like to congratulate the authors on this work!

Response:

We are extremely grateful to the Respected Reviewer for the insightful comments and suggestions. These comments and suggestions have enabled the substantial improvement of the quality of the manuscript.